# Hub stability in the calcium calmodulin-dependent protein kinase II
Chih-Ta Chien [1,6], Henry Puhl[2], Steven S. Vogel [2], Justin E. Molloy [3,7], Wah Chiu [1,4] ✉ & Shahid Khan [5] ✉

The calcium calmodulin protein kinase II (CaMKII) is a multi-subunit ring assembly with a central hub formed by the association domains. There is evidence for hub polymorphism between and within CaMKII isoforms, but the link between polymorphism and subunit exchange has not been resolved. Here, we present near-atomic resolution cryogenic electron microscopy (cryo-EM) structures revealing that hubs from the α and β isoforms, either standalone or within an β holoenzyme, coexist as 12 and 14 subunit assemblies. Single-molecule fluorescence microscopy of Venus-tagged holoenzymes detects intermediate assemblies and progressive dimer loss due to intrinsic holoenzyme lability, and holoenzyme disassembly into dimers upon mutagenesis of a conserved inter-domain contact. Molecular dynamics (MD) simulations show the flexibility of 4-subunit precursors, extracted in-silico from the β hub polymorphs, encompassing the curvature of both polymorphs. The MD explains how an open hub structure also obtained from the β holoenzyme sample could be created by dimer loss and analysis of its cryo-EM dataset reveals how the gap could open further. An assembly model, considering dimer concentration dependence and strain differences between polymorphs, proposes a mechanism for intrinsic hub lability to fine-tune the stoichiometry of αβ heterooligomers for their dynamic localization within synapses in neurons.

The calcium calmodulin-dependent kinase II (CaMKII) is unique among the calmodulin (CAM) kinase superfamily (~80 members) in that it forms a multi-subunit ring[1,2]. There are four major isoforms (α, β, γ, δ) in mammals. The α and, to a lesser extent, β isoforms are predominantly expressed in the brain, while δ is the major isoform in the heart[3,4]. Subunit architecture has a common domain organization across species and isoforms (Fig. 1a). The association domain (AD) forms the central hub. Linkers with variable length and sequence between isoforms and developmental splice variants[3], connect each AD with a peripheral kinase domain (KD) (Fig. 1b). The KD, in addition to canonical N and C-terminal lobes, contains a regulatory, pseudo-substrate α-helix (R) with the activating T287 and inhibitory (T306-307) autophosphorylation sites. The AD is part of a large superfamily with a β sheet anchored to a long α-helix as its signature fold. The AD hub, which evolved from bacterial dimeric enzymes, extends the β sheet to form lateral contact interfaces (LCs) needed for ring assembly, while the ancient dimer contact was preserved as the vertical contact interface (VC) in the mirror-

symmetric two-stack hub[5]. The VC dimer will henceforth be referred to as the "dimer".

In the inactive state, R is docked to the KD core substrate site. Ca$^{2+}$.CAM binding undocks R and enables the trans-autophosphorylation of T287 on the undocked R by an adjacent KD for kinase activation[1]. The undocked R can also interact with the hub, modulated by T306-307 autophosphorylation[6]. In vitro, CaMKII kinase activity depends on the frequency of Ca$^{2+}$ oscillations[7], consistent with Ca$^{2+}$ decoding functions central to mammalian physiology[8].

High-resolution hub structures obtained by X-ray crystallography for the mammalian isoforms (α, β, γ, δ) have revealed hubs trapped in different oligomeric states depending on isoform[9-11], engineered residue substitutions[12], or the presence of the KD[13,14]. Hub dynamics are of interest since changes in hub composition due to differences in isoform or mutation affect kinase activity[15]. In addition, it has been shown that kinase activation triggers subunit exchange[16]. Molecular dynamics (MD) simulations of

[1]Department of Bioengineering, and Department of Microbiology and Immunology, James H. Clark Center, Stanford University, Stanford, CA 94305, USA. [2]Laboratory of Biophotonics and Quantum Biology, National Institutes on Alcohol, Abuse and Alcoholism, National Institutes of Health, Rockville, MD 208952, USA. [3]The Francis Crick Institute, London, UK. [4]CryoEM and Bioimaging Division, Stanford Synchrotron Radiation Light source, SLAC National Accelerator Laboratory, Stanford University, Menlo Park, CA 94025, USA. [5]Molecular Biology Consortium @ Lawrence Berkeley National Laboratory, Berkeley, CA 94720, USA. [6]Present address: Laboratory of Molecular Biology, National Institute of Diabetes and Digestive and Kidney Diseases, National Institute of Health, Bethesda, MD 20892, USA[7]Present address: CMCB, Warwick Medical School, Coventry CV4 7AL, UK. ✉e-mail: wahc@stanford.edu; smkhan@lbl.gov

## a

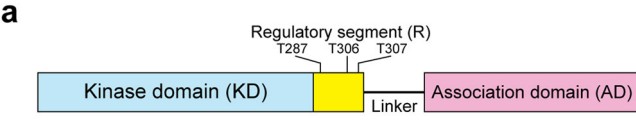

## b

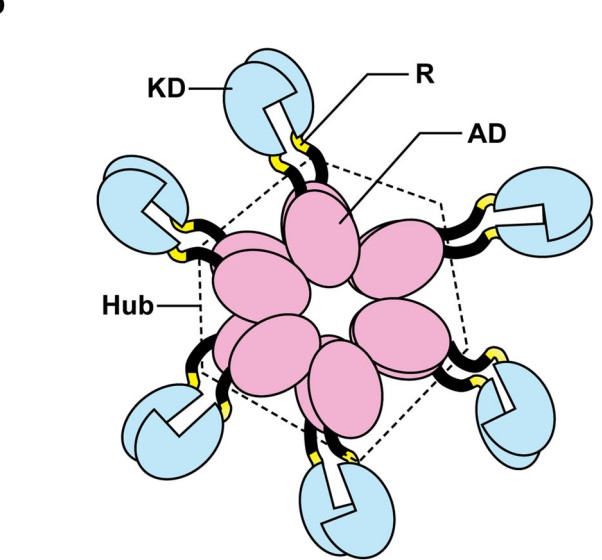

**Fig. 1 | Schematic of the CaMKII holoenzyme subunit composition and organization. a** The subunit is composed of a kinase domain (KD in blue), a regulatory segment (R in yellow), a linker (in black), and an association domain (AD in pink). **b** The subunit architecture of the CaMKIIβ holoenzyme. The AD forms the central hub (dashed hexagon). The peripheral KDs radiate out from the hub connected by flexible linkers that vary between isoforms. The regulatory segment, R, can alternate between multiple conformations within a single subunit. Alanine mutagenesis silences the activating (T287) and inhibitory (T306, T307) autophosphorylation sites.

crystal structures supplemented by single-molecule total internal reflection fluorescence microscopy (TIRFM) have deciphered the mechanism of kinase activation-triggered subunit exchange, but these investigators also noted that α holoenzymes disassemble without kinase activation as evidenced by a coexisting pool of dimers in the heterogenous holoenzyme populations[6,17]. We will refer to this activation-independent disassembly as "intrinsic lability". Evidence for subunit polymorphism within hubs for the mammalian α and β isoforms has been based on 2D-class averages of top-view projections from negative stain EM[18,19] and cryo-EM images[15]. One study[18] reported a one-to-one dodecamer/tetradecamer ratio in a 5 μg/ml sample by electrospray ionization mass spectrometry (ESI-MS). The cryo-EM study[15] also presented 3D reconstructions of the hub from an α holoenzyme sample, but the residue contacts were not resolved even at the best-reported resolution (4.8 Å).

Here, we present cryo-EM structures of the isolated rat α and β hubs at a resolution comparable to the crystal structures. Near atomic resolution structures of different oligomeric states were resolved from each sample, demonstrating the ability of single-particle cryo-EM to sort out heterogeneous populations. Analysis of the resolvability of AD contacts, assessed by the Q-score[20], coupled with MD simulations, reveals the basis for the subunit polymorphism reported by the earlier EM studies. The analysis was supported by single-site mutagenesis of a critical phenylalanine in the LC. The intact β holoenzyme had greater hub lability and heterogeneity, in line with reports for the human α holoenzyme[21]. In addition to the polymorphic hub structures, we obtained an open hub structure lacking one dimer from the β holoenzyme sample. The dimer has been proposed to be the fundamental unit for subunit exchange[16]. Our current 3D structures generate the first direct visual evidence of this hypothesis. Furthermore, we applied three-dimensional flexible refinement (3DFlex)[22], a recently developed

methodology for analyzing continuous motion in cryo-EM data, supplemented with MD simulations to characterize open hub flexibility and support the proposal for serial hub assembly/disassembly by dimer association/dissociation[18]. The MD revealed that the open hub relaxes to open further within 100 nanoseconds in the absence of a dimer pool. Parallel TIRFM experiments showed that phosphorylation-silent holoenzymes ($β_{T287.306-307A}$) dissociate into smaller assemblies down to dimers within minutes at nanomolar concentrations. We suggest that the polymorphism is due to the balance between dimer concentration-dependent serial assembly and ring closure isomerization. In neurons, the intrinsic lability would allow the composition of CaMKII αβ hetero-oligomers to be adjusted to variations in isoform levels[23].

## Results

### Variable symmetry in isolated CaMKII α and β hubs

We employed cryo-EM to investigate the symmetry variation in CaMKII hubs because of their ability to resolve multiple structures in a single sample. We expressed and purified a recombinant form of CaMKII comprising the α AD fused to a fluorescent Venus tag, separated by a flexible 15-amino-acid linker (Fig. 2a). The Venus tag allowed the lability to be tracked in gel-filtration and single-molecule TIRFM experiments. From a single sample, we obtained two cryo-EM structures for the CaMKIIα hub (Fig. 2b), a dodecamer (12-mer) and a tetradecamer (14-mer), with 6 and 7 ADs in each of the two stacked rings, respectively. The Venus tags were observed as a blurry density around the hub in 2D averages for both 12-mer and 14-mer (Supplementary Fig. 1b, c). The 12-mer and 14-mer hubs were resolved to 2.7 and 2.6 Å, respectively. The 14-mer hub has a 12 Å increase in diameter and a larger central pore compared to the 12-mer hub, as seen previously in negative-stain single particle 2D-class averages of holoenzyme hubs[18,19]. The hub has two distinct inter-domain contacts, LC and VC, as a consequence of its two-stack mirror symmetry.

Cryo-EM maps of 12-mer (2.6 Å resolution) and 14-mer (2.6 Å resolution) were also obtained for the β-isoform (Fig. 3). The superimposed structures of the 12-mers between α-isoform and β-isoform had a root mean square deviation (RMSD) of 0.35 Å, while that for the comparison of the 14-mers between the two isoforms is 0.3 Å. We conclude that polymorphism is an intrinsic property of the α and β hubs, independent of the KD. Metrics of map quality and local resolution maps for both hubs are presented in Supplementary Fig. 1 (α hub) and Supplementary Fig. 2 (β hub), respectively.

Atomic models of the 12-mer and 14-mer hubs were built from the maps (Fig. 2c). The overlay between the cryo-EM map and the atomic model for a single AD from the 14-mer α hub, shows a well-resolved backbone, except for two loops plus the N- and C-termini. The average Q-score for secondary structure elements, 0.73 for α-helices and 0.85 for β-sheets are higher than that for loops (0.66). These scores are in line with expectations since loops are typically flexible, and the extended β-sheet anchors the AD fold.

Next, we mapped the resolvability of the VC and LC residues in terms of Q-score[20]. The VC is notable for the abundance of histidine residues (Figs. 2d and 3). The homologous contact extends across the β-sheets with the same set of residues involved from each AD with 32.2° rotation between them. It has a predominantly polar character, due in major part to histidines. The histidines, H420, H466, and H468, form hydrogen bonds with T378, T452, and S470. On the other hand, the LC occurs between α-helices α3 and α4 of one AD (*) with β-strands β3–β5 of the adjacent AD (#) (Fig. 2c, e). The F394* backbone forms a hydrogen bond with the Q436# side chain (Fig. 2e). Two hydrophobic patches above (F397, L438, P444) and below (L385, F394, I434, L415, V389) the F394–Q436 hydrogen bond provide additional stabilization. The α LC has been characterized by site-specific alanine mutagenesis of 10 residues that, ranked by efficacy based on disassembly, were F394* > Q436# > I434# > F397*. Residues D393*, Y398*, H411#, T413#, and L415# had no effect[24]. These contact residues are conserved between the two isoforms with the exception that αI434 (Fig. 2e) is changed to βL498 (Fig. 3).

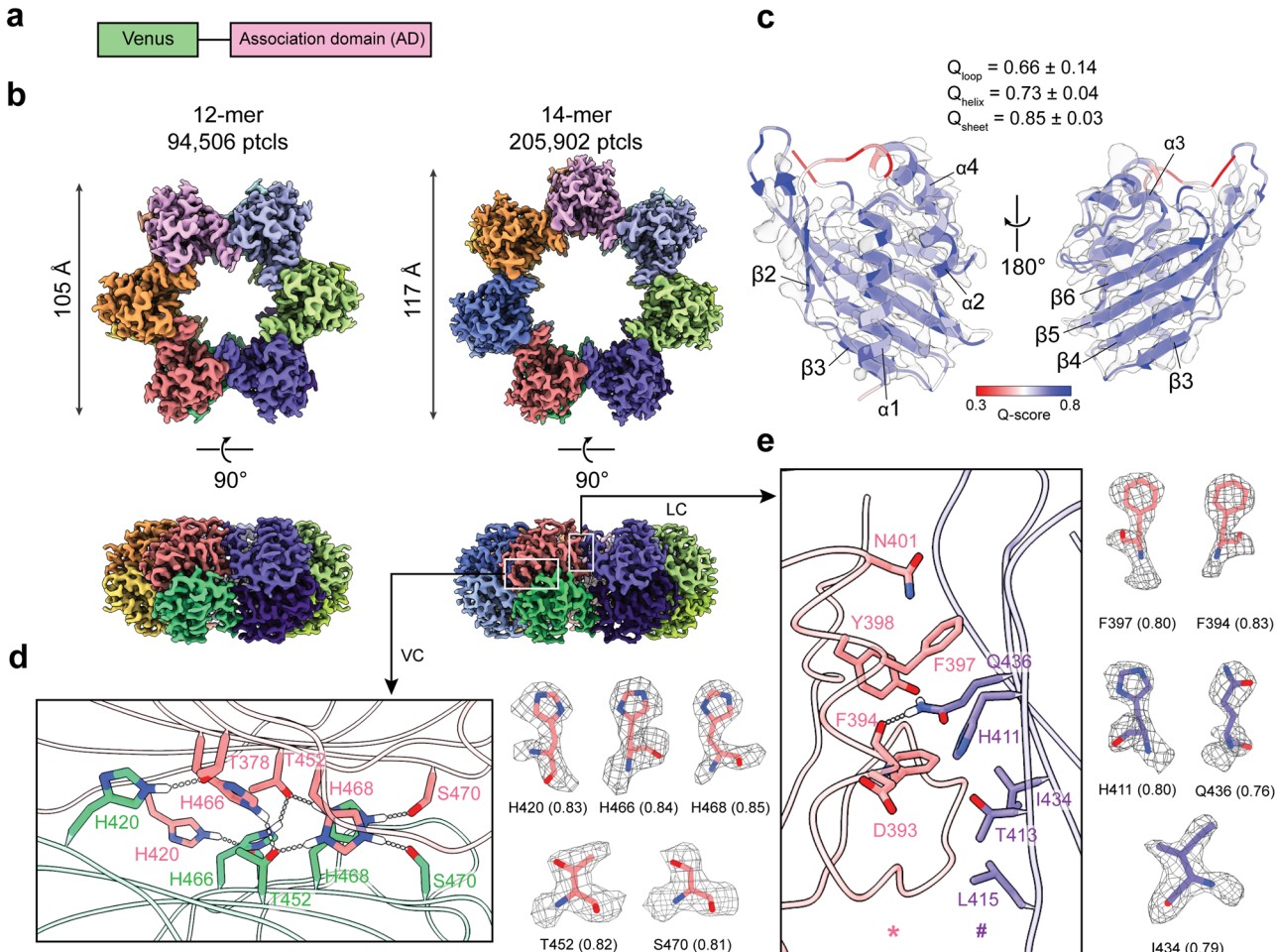

**Fig. 2 | Cryo-EM structures of CaMKIIα hubs. a** Diagram of the domains of the CaMKIIα hub construct used in this study. **b** Top view and side view of the cryo-EM maps of the 12-mer (left) and 14-mer (right) AD hubs from the CaMKIIα construct particle numbers (ptcls) used for the map reconstructions are listed. The lateral contact interface and vertical contact interface are indicated with white boxes. **c** Cryo-EM density and molecular model overlay for the single CaMKIIα AD. The model was colored by Q-score from 0.3 (red) to 0.8 (blue). Average Q-scores for each secondary structure element and for loops are indicated. **d** Key residues in the vertical contact interface, consisting of a histidine hydrogen bond network. **e** The lateral contact interface consists of a critical F394–Q436 hydrogen bond and two hydrophobic patches. The two ADs denoted as * and # are colored in pink and purple, respectively. Cryo-EM density and model overlays for the key residues are shown in **d** and **e**, with the Q-score for each residue indicated in the brackets.

We extended the comparison to all hub structures with bioinformatic measures. While 16-mer hubs have been visualized[19], published high-resolution (<3 Å resolution) hub structures are of 14-mers or 12-mers, with mammalian α hubs being the most abundant. The sequences from the structures were aligned, and the residues conserved across the mammalian isoforms were mapped onto our 14-mer cryo-EM α hub to show the dielectric variations in the contact interfaces (Supplementary Fig. 3a–d). The VC and LC energetics, computed with ePISA[25], were remarkably similar across all the hub structures, including our cryo-EM structures, showing the LC was less stable than the VC (Supplementary Fig. 3e).

The ePISA analysis suggested that the F–Q hydrogen bond is the key contributor to LC stabilization ubiquitous to all hub structures (Supplementary Fig. 3f). The bond reduces the energy cost for the burial of the polar Q sidechain within the largely hydrophobic contact in addition to solvation of the aromatic F sidechain. We mutated residue βF458, corresponding to αF394, to alanine to validate the ubiquitous role of the F–Q hydrogen bond in hub stability. Gel-filtration results (Supplementary Fig. 3g) showed that the βF458A sample contains mostly small assemblies, whereas the major population in the native β eluted between 10 and 12 mL, consistent with 14-mer/12-mer assemblies. These results confirmed that the F458A residue substitution disassembles isolated β hubs.

## The modulation of map resolvability and flexure with assembly state

We employed molecular dynamics (MD) simulations to supplement the Q-score and ePISA measurements for insight into the dynamics and flexibility of the inter-domain contacts of the 14-mer and 12-mer polymorphs. The tetramer formed by the association of two dimers is the smallest precursor assembly with both VC and LC interfaces. We reasoned that the tetramer contact dynamics and angular curvature in the absence of steric constraints due to ring formation would give insight into intrinsic flexibility. In parallel, per residue Q-scores provide an independent measure as structural flexibility is one of the major factors determining cryo-EM map resolvability. We performed 50 ns MD simulations, in triplicate, of tetramer structures extracted in-silico from the cryo-EM hub structures. The root mean square fluctuation (RMSF) values showed that the VC (Fig. 4a) is more rigid than the LC (Fig. 4b) in the tetramer extracted from the α 14-mer ($\alpha_{14T}$), consistent with the ePISA estimates of the energetics. The result is also consistent with the published simulations[16]. The lower RMSF values of VC correlated with higher Q-score (Fig. 4c). The negative correlation between Q-scores and the $C_\alpha$ RMSF over the α hub had a Pearson coefficient of −0.6. We focused on the less well-characterized β hub, having validated our simulation approach with the α hub. Parallel simulations on tetramers extracted from the β 12-mer ($\beta_{12T}$) and 14-mer ($\beta_{14T}$) hubs gave mean

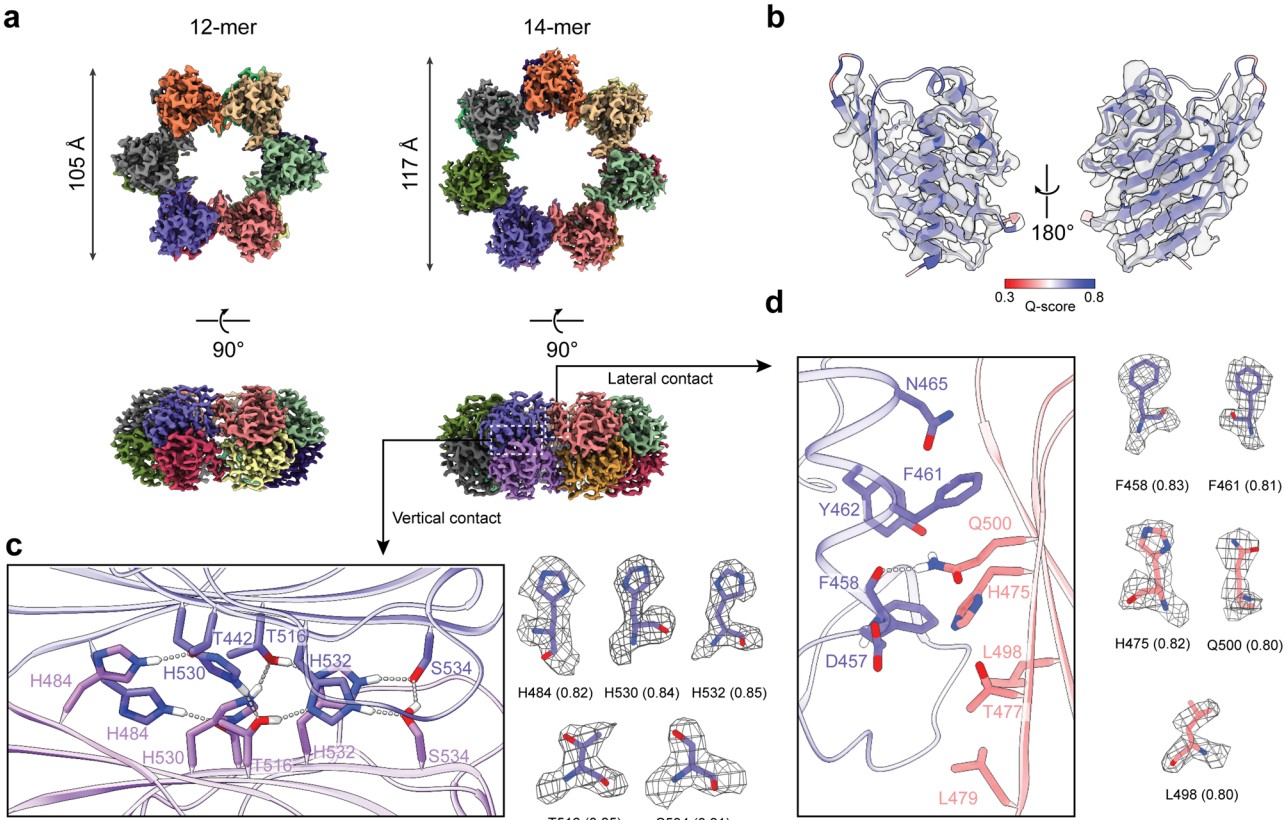

**Fig. 3 | Cryo-EM structures of CaMKIIβ hubs. a** Top view and side view of the cryo-EM maps of 12-mer (left) and 14-mer (right) CaMKIIβ hubs. Lateral contact and vertical contact are indicated with white dashed boxes. **b** Cryo-EM density and molecular model overlay for the single CaMKIIβ AD. The model was colored by Q-score from 0.3 (red) to 0.8 (blue). **c** Key residues involved in the vertical contact, which consists of a histidine hydrogen-bond network. **d** The lateral contact consists of a critical F394-Q436 hydrogen bond and two hydrophobic patches. Cryo-EM density and model overlays for the key residues are shown in **c** and **d**, with the Q-score for each residue indicated in the brackets.

RMSF values ($\beta_{12T} = 0.11 \pm 0.02$, $\beta_{14T} = 0.11 \pm 0.01$) comparable to $\alpha_{14T}$ (0.09 ± 0.03). The overall Q-score/RMSF Pearson's correlation coefficients ($P_{\beta12} = -0.54$, $P_{\beta14} = -0.45$) were lower. The Q-score anti-correlations were more pronounced ($P_{\alpha14} = -0.63$, $P_{\beta12} = -0.68$, $P_{\beta14} = -0.62$) when compared with RMSF-derived isotropic residue B-factors ($= (8/3)\pi^2 RMSF^2$).

We defined two angles $\gamma$ and $\delta$ as a measurement of the curvature to compare the flexibility of the extracted tetramers with the parent β hubs (Fig. 4d). The angle $\gamma$, defined as the angle between F458 (chain A), L498 (chain A), and F458 (chain B), measures the radial curvature along one stack perpendicular to the central axis. The $\gamma$ is primarily sensitive to VC fluctuations based on the MD simulation. The angle $\delta$, defined as the angle between the first vector, R410 and G428, in chain A, and the second vector, R410 and G428, in chain B. $\delta$ measures the divergence between the long α1 helices of the adjacent ADs angled roughly 35° from the central axis. The $\delta$ is primarily sensitive to LC fluctuations. The tetramer $\gamma$ angular distribution has a modestly shifted mean and greater spread than the parent hub, as illustrated for the 14-mer (Fig. 4e). In contrast, the tetramer $\delta$ angular distributions extracted from the 12-mer and 14-mer hubs have substantially broadened distributions relative to the narrow distributions of the 12-mer and 14-mer parent hubs (Fig. 4f). The $\delta$ angular distributions for the parent hubs are well-resolved from each other, whereas the tetramer distributions were superimposable with a mean $\delta$ angle intermediate between the mean values for the parent hubs. The $\delta$ and $\gamma$ angle measurements reveal the constraints, particularly to the LC, imposed by ring formation. Importantly, they suggest that the 12-mer and 14-mer have comparable stability as their $\delta$ angle curvature deviates slightly higher or lower, respectively, from the most stable $\delta$ angle determined from the unconstrained tetramer sub-assemblies. The simulations of their dynamics explain why coexisting 14-mer and 12-mer CaMKIIα holoenzyme assemblies have been repeatedly visualized in the previous EM studies based on 2D image analysis[15,18,19].

## Heterogeneity in CaMKIIβ holoenzyme populations

Differences have been documented between isolated hubs versus those in intact holoenzymes due to KD-AD interactions. In order to determine activation-independent effects, we engineered a CaMKIIβ holoenzyme construct with alanine substitutions at both activation (T287) and inhibitory (T306.T307) phosphorylation sites, the $\beta_{T287.306-307A}$ holoenzyme. The T287A substitution blocks activation and aids R attachment to the KD core, while the T306-307A substitutions countered the dissociation reported for human α hubs by the addition of R peptides phosphorylated at these sites[6].

We first checked that the parent strain mediates calcium/calmodulin-stimulated T287 autophosphorylation (Supplementary Fig. 4a) and compared the stability of the $\beta_{T287.306-307A}$ holoenzyme (Supplementary Fig. 4b) with the isolated β hub (Supplementary Fig. 3g) using gel-filtration. The F458A residue substitution generated a major population of dimers in both holoenzymes and isolated hubs. Superimposition of the $\beta_{T287.306-307A}$ and $\beta_{F458A}$ profiles revealed a peak in the $\beta_{T287.306-307A}$ profile with an elution volume of 9 ml, expected for intact holoenzymes, but also peaks at the dimer and monomer positions identified by the $\beta_{F458A}$ profile. Thus, the rat β holoenzyme is markedly less stable than the isolated hub, consistent with the human α holoenzyme[21]. We conclude that the silencing T287A and T306-307A residue substitutions may reduce, but not eliminate, hub disassembly.

The gel-filtration documented that the holoenzyme sample is more labile and heterogeneous than the isolated hubs but did not inform on disassembly kinetics or mode of disassembly. We turned to single-molecule, real-time TIRFM to answer these issues. The assembly states in purified

**Fig. 4 | RMSF and *Q*-score correlation.** Tetramer sub-assemblies ($\alpha_{14T}$) were extracted from CaMKIIα 14-mer hub structure in-silico, and MD simulated for 50 ns. The resulting RMSF values for **a** VC and **b** LC are mapped to Cα atoms shown in spheres. **c** RMSF (red lines, left *y*-axis) and *Q*-scores (black lines, right *y*-axis) for CaMKIIα 14-mer hub. The RMSFs were obtained from simulations of the tetramer sub-assembly. The *Q*-scores were averaged over the complete hub. The VC and LC are indicated by blue and pink areas, respectively. *P* Pearson correlation coefficient. **d** Definition of angles $\gamma$ and $\delta$. $\gamma$ (orange line) is the angle between F458 (chain A), I498 (chain A), and F458 (chain B). $\delta$ (blue lines) is the angle between the first vector, R410 and G428, in chain A, and the second vector, R410 and G428, in chain B. **e** Decrease in $\gamma$ curvature angle and spread upon transition from the extracted β tetramer ($\beta_{14T}$, squares (mean ± SE)) to the 14-mer parent hub (circles (mean ± SE)). **f** The $\delta$ spread in the β tetramer sub-assemblies ($\beta_{12T}$, $\beta_{14T}$ (mean ± SEs)) is reduced and resolved into distinct 14-mer and 12-mer distributions by ring formation.

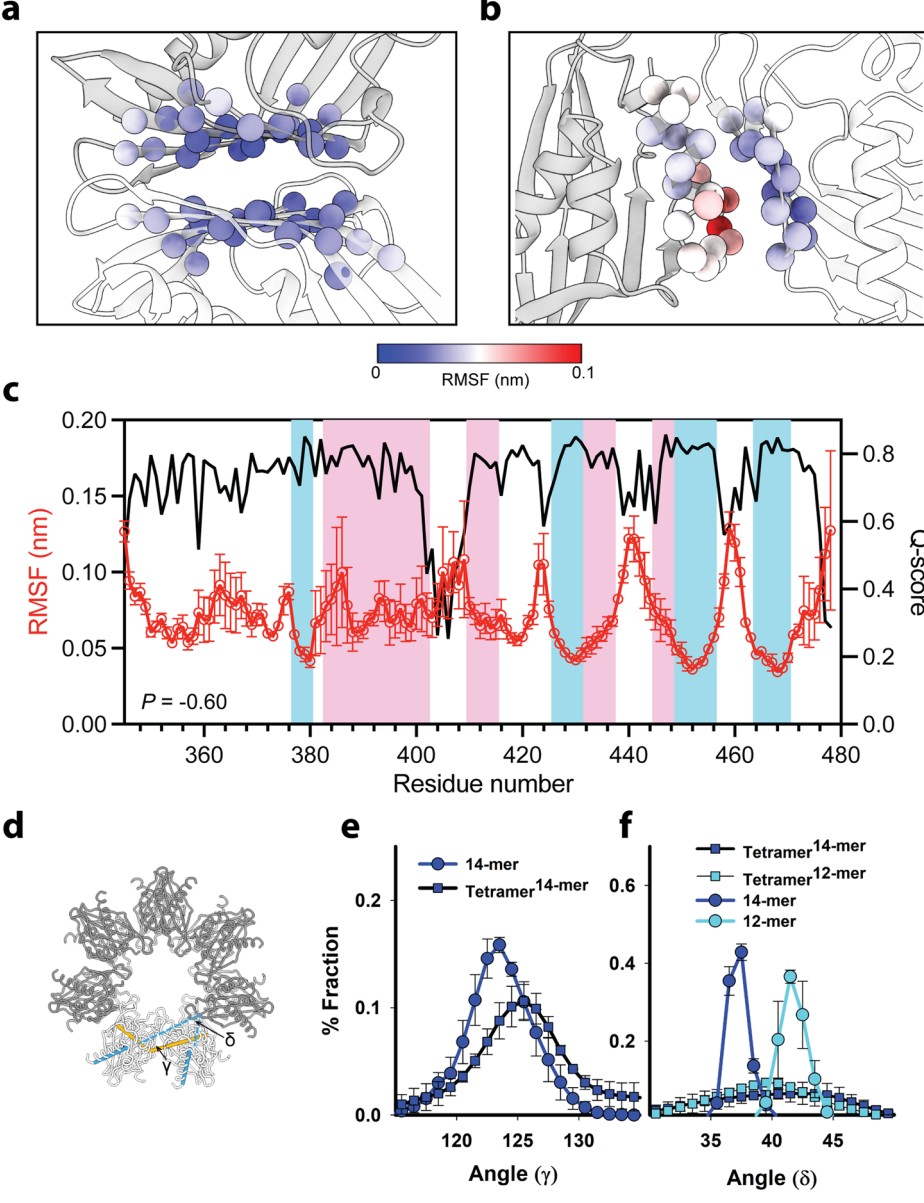

Venus-tagged CaMKII samples, immobilized on a GFP antibody-coated surface, were determined by intensity analysis and spot photobleaching. The intensity distributions of single spots from immobilized $\beta_{F458A}$ and $\alpha_{F394A}$ populations were bimodal and indistinguishable from each other, while the heterogeneous $\beta_{T287.306-307A}$ and native holoenzymes distributions were shifted towards higher intensities (Fig. 5a). The photobleaching of single spot intensities was analyzed with a custom step-finder algorithm for the estimation of assembly subunit stoichiometry. Video records revealed stepwise photobleaching with the mean of the Gaussian $\beta_{F458A}$ photobleaching step-size intensity distribution comparable to the unitary Venus fluorescence, while the step-size in the heterogeneous distributions of β and $\beta_{T287.306-307A}$ was skewed towards somewhat higher values due to methodological limitations (Supplementary Fig. 4c–e). The comparison of the $\beta_{T287.306-307A}$ stoichiometry distribution between experimental records taken at two-time points ($t = 15$, 45 minutes) after 1000× dilution of the stock revealed that the dimer fraction increased with time, with a concurrent decrease in the larger assemblies (>8-subunits) (Fig. 5b). The shift was not observed when stocks were incubated for 45 min at ambient temperature then diluted 1000x fold for observation, indicating that larger assemblies are maintained by storage at high concentration. The separation of the

immobilized $\beta_{T287.306-307A}$ holoenzymes from unattached $\beta_{T287.306-307A}$ complexes by buffer washout during slide preparation would remove the pool responsible for such maintenance. In gel filtration, the separation is a natural, albeit more gradual, consequence of the size exclusion chromatogram. A "dark" fluorophore fraction[26] would not explain the observed shift in the distribution towards dimers, but it may contribute to the heterogeneity. The TIRFM experiments, supported by the gel filtration, demonstrated that $\beta_{T287.306-307A}$ holoenzymes disassemble within minutes into smaller assemblies due to dimer loss.

## Cryo-EM structures and flexibility analysis of holoenzyme hubs
We obtained three cryo-EM reconstructions from the engineered β holoenzyme sample: a 14-mer (3.0 Å resolution), a 12-mer open ring (3.5 Å resolution), and a close 12-mer (8.4 Å resolution) (Fig. 6). The close 12-mer cryo-EM reconstruction has a much lower resolution, suggesting its structural heterogeneity. Metrics of map quality and local resolution maps for the 14-mer and 12-mer open hubs are presented in Supplementary Fig. 5. We were only able to resolve the hubs in these structures, indicating that Venus-tagged KDs, separated from the ADs by the flexible 93-residue linker[27], are not aligned (Fig. 6a). No additional density that might be attributed to the

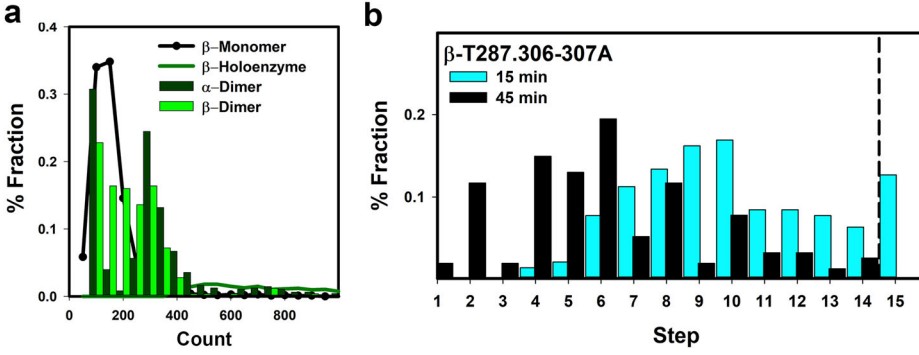

**Fig. 5 | Single-molecule TIRFM.** Duplicate experiments (>20 records/experiment), with typically 30–50 immobilized V-CaMKII spots per record, were conducted for each condition (>1000 spots). Spot intensities were averaged over 20 s (100 frames). **a** The $\alpha_{F394A}$ and $\beta_{F458A}$ bimodal intensity distributions are relative to the βmonomer and βholoenzyme distributions. The higher mode intensity was double that of the lower mode. The latter superimposed with the mode of the βmonomer distribution ($I_{UNI}$) The $\beta_{F458A}$ population fraction with intensity values > 4*$I_{UNI}$

was <10%. The F-test statistic ($F_{obs}/F_{pred} = 1.22/1.84 < 1$) indicates that the variances of the $\alpha_{F394A}$ and $\beta_{F458A}$ distributions do not differ (probability $p > 0.05$). **b** Time-dependent room temperature disassembly in $\beta_{T287.306-307A}$ holoenzyme populations revealed by comparison of measurements on 30 spots from different slides observed 15 (409 steps) and 45 (384 steps) minutes after dilution from the stock on ice. The F-test statistic ($F_{obs}/F_{pred} = 0.97/0.39 > 1$) indicates there is a significant difference between the two populations ($p < 0.05$).

**Fig. 6 | The open ring structure of CaMKIIβ holoenzyme. a** Diagram of the domains of CaMKIIβ holoenzyme constructs used in this study. Ptcls particles. **b** Top and side views of three cryo-EM maps of AD hubs derived from a single CaMKIIβ sample, a 14-mer, an open 12-mer, and a close 12-mer at 3.0, 3.5, and 8.4 Å, respectively. The open 12-mer map was sharpened using DeepEMhancer[58] due to its varying local resolution. 2D class averages of the top views are shown. The side view of the open ring 12-mer shows a 14.6° off-plane shift of the right-end vertical dimer. Supplementary Video 1 shows the transition from the 14-mer structure to the 12-mer open structure.

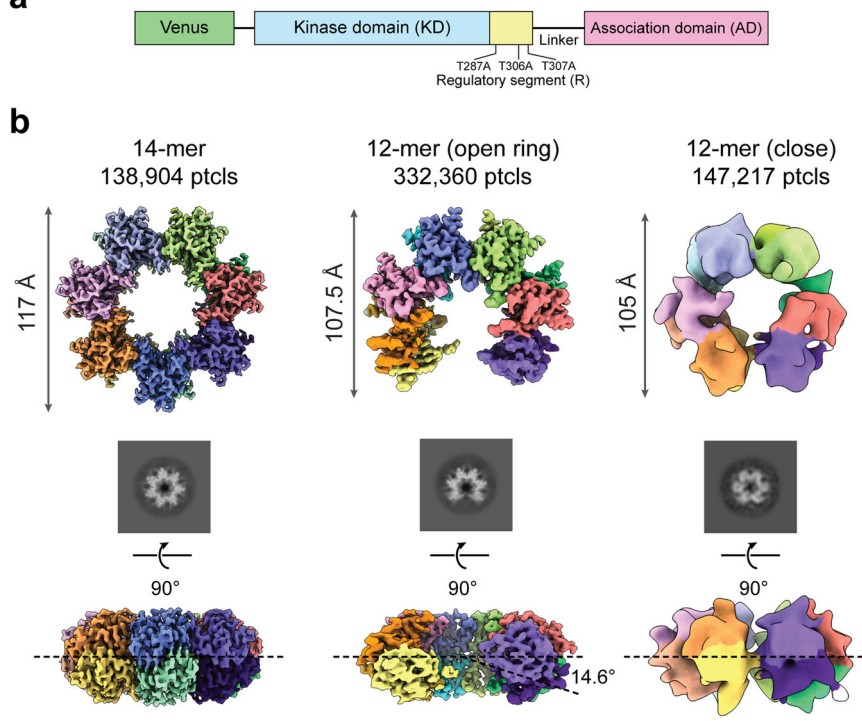

tagged KDs was detected in the 2D-class averages. The 14-mer hub structures from the β holoenzyme and the isolated β hub superimposed with an RMSD of 0.52 Å. This result implies that the dominant population is composed of inactive holoenzymes in the extended state[28]. The open ring structure is the first near-atomic resolution structure of an intermediate CaMKII hub assembly. It has 12 ADs but a diameter similar to the 14-mer ring. The ADs adjacent to the opening are blurred in the top view 2D class averages (Fig. 6b) with lower local resolutions (Supplementary Fig. 5g). The vertical dimer on the right of the opening shifts 14.6° down off-plane (Fig. 6b). A morph movie (Supplementary Video 1) was generated for easier comparison between the 14-mer and the open 12-mer. The transition from the 14-mer to the open 12-mer can be explained by a progressive relaxation of the AD extended β-sheet as illustrated in the video by superposition of the ADs from the corresponding atomic models.

The poor resolvability of the ADs adjacent to the gap was documented by the Q-scores (Fig. 7a). We utilized the 3D-flex refinement algorithm for analysis of the motion in the open 12-mer cryo-EM dataset (Fig. 7b, c). The analysis indicated that the dominant collective motions are the enlargement of the gap by 5–10 Å due to weakened LCs between the adjacent ADs (Supplementary Video 2). These motions largely explain the lower resolvability of these ADs.

Finally, we employed MD simulations to provide a chemical rationale for the dynamics revealed by the Q-scores and 3D-flex analysis for the open 12-mer (Supplementary Fig. 6). We constructed an open ring hub in-silico by extraction of a vertical dimer from the β 14-mer hub, We simulated this assembly for 100 ns following analogous simulations reported for the human 12-mer γ hub[16]. Increased disorder, as assessed by the RMSF, of the ADs adjacent to the gap within 10 ns of simulation spread to more distant

**Fig. 7 | The dynamics of the 12-mer open structure extracted from cryo-EM data. a** $Q$-score derived from the open ring cryo-EM map and model. $C_\alpha$ atoms are color-coded by $Q$-score. **b** Distribution of the particles in the 2D latent space from the 3D-flex analysis[22] of the open 12-mer cryo-EM dataset. Each blue dot is one of the 332,360 particles. **c** Start (blue) and end images (red) and the overlap of the two from particle reconstructions along the latent coordinate 1 and latent coordinate 2 trajectories. The motions are shown in Supplementary Video 2.

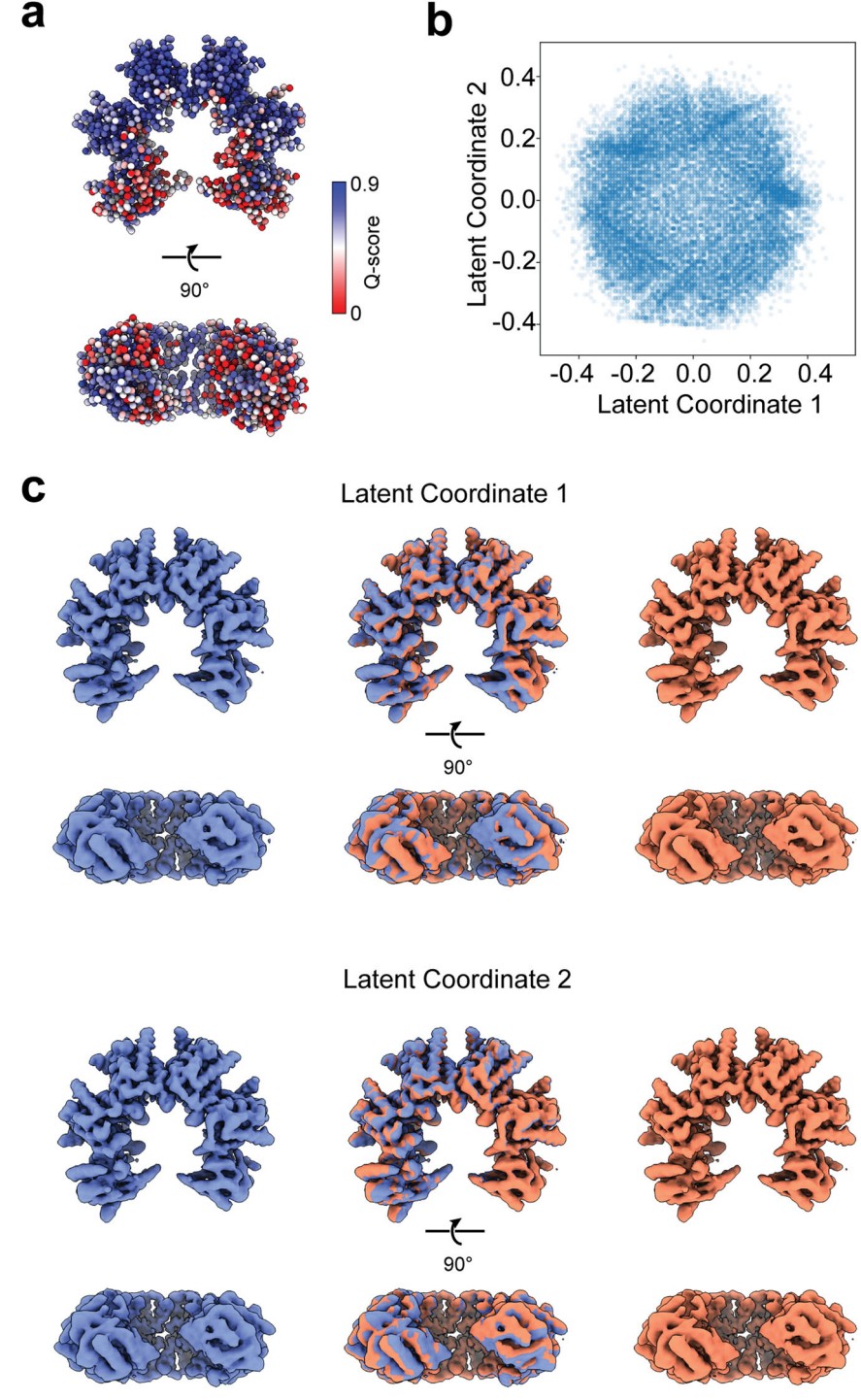

ADs along both stacks with increased run time. Other initiation sites were not seeded over the 100 ns simulation runs. We defined an angle $\varphi$ to measure AD separation across the gap. The angle $\varphi$ was stable for control simulations on the closed hubs (20 ns) (Supplementary Fig. 6a–c). It increased with time in simulations of the open hub with episodic jumps interspersed with more gradual increases. Any corresponding changes in the root mean square deviation (RMSD) of the ADs adjacent to the gap were small and uncorrelated with the episodic $\varphi$ jumps (Supplementary Fig. 6d–g). 100 ns is too short for $\varphi$ to reach a stationary value, as discussed previously by others[16], but the trend is that once a dimer dissociates, LCs adjacent to the dissociation site progressively weaken to further open the hub, consistent with the 3D-flex analysis. We speculate that in the absence of

a dimer pool, as obtained in the TIRF experiments, this could lead to further dimer dissociation, but much longer simulations will be required to test this proposition.

## Discussion

Previous studies have noted that the central hub of the CaMKII holoenzyme is conformationally strained and likely to be intrinsically labile. A major effort has focused on the functional role of the activation-triggered subunit exchange[8]. Here, we have studied isolated hubs of the two neuronal isoforms (α, β) together with an engineered phosphorylation silent β holoenzyme to separate intrinsic liability from that caused by interactions of the activated kinase domain with the holoenzyme hub. Previous studies[15,18,19] have shown

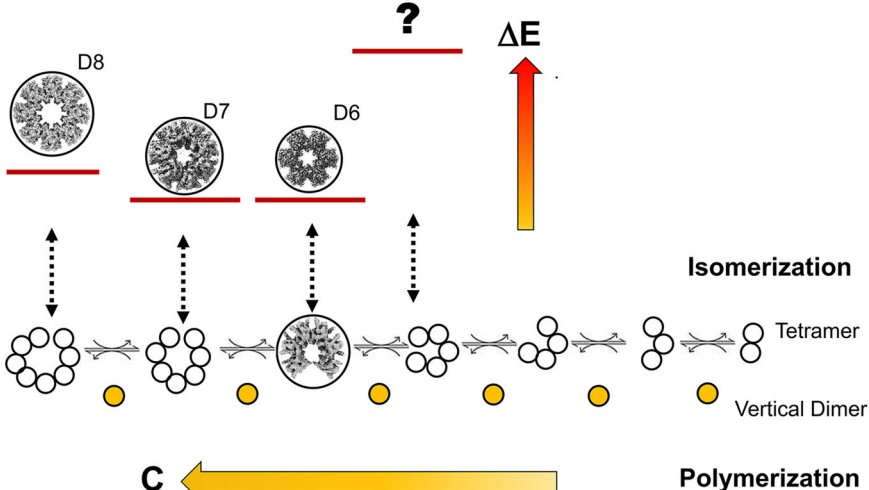

**Fig. 8 | CaMKII hub assembly model.** The assembly of polymorphic hub structures from the tetramer precursor structure entails a balance between polymerization and isomerization. The polymerization of progressively larger open structures depends on dimer concentration (*C*), analogous to the concentration dependence of the open, spiral metazoan hub. The free energy (Δ*E*) for hub formation depends on the trade-off between the strain in the closed hub and the solvation of the four LCs upon dimer addition. The axes for both (*C*) and (Δ*E*) are relative. The open hub resolved in this study would require concentration-dependent reinsertion followed by ring closure to obtain the 14-mer hub, or it may close to form the 12-mer hub. The 12-mer and 14-mer hubs are the dominant forms seen in all EM studies thus far. The deviation of these structures from the intrinsic curvature found in the tetramer has the opposite sign but equal amplitude. For replica and negative-stain samples where particle concentrations are typically lower than for cryo-EM, the 12-mer hub might be the favored form. Freeze-etch replicas have reported a β 10-mer[59], but the stoichiometry was estimated from the peripheral KDs that may have been lost by proteolysis of the long β KD-AD linker. A β 16-mer has been crystallized, but EM suggests that it is a minor species (<0.4%) relative to the 12-mer (92.7%) at the concentration used in a negative stain study[19]. The 16-mer and the 10-mer (depicted as ? to note its existence is questionable), will be more strained than the dominant polymorphs due to the greater deviation of their curvature from the intrinsic LC curvature. An even greater penalty might prohibit the formation of smaller hubs, and none have been reported.

that hub polymorphs coexist in holoenzyme preparations. Our study shows that polymorphism is also present in biochemically purified α and β hubs (Figs. 2, 3) as well as the β holoenzyme (Fig. 6). Our near-atomic resolution hub structures allowed us to correlate the resolvability of the inter-domain contact residues with atomistic MD simulation of their dynamics (Fig. 4). The disassembly of the β hub by alanine mutation of the critical LC phenylalanine (F458) is consistent with the previously observed result for the α holoenzyme[24] and validates the conservation of the lateral contact based on the sequence identity and energetics (Supplementary Fig. 3).

The native β and $β_{T287,306-307A}$ holoenzyme populations are more heterogeneous than the isolated β hubs, consistent with the previous calorimetric measurements that isolated hubs are more stable than the holoenzyme of the human α isoform[21]. Structures of the KD were not visualized in our holoenzyme structures, given the long β KD–AD linker. Nevertheless, our TIRFM data imply that the KD interacts with, and disassembles $β_{T287,306-307A}$ hubs within minutes in the absence of a pool of free subunits. The proposal that the vertical dimer is the fundamental unit of assembly has been substantiated since its inception[16] by various lines of evidence. We have presented here a near-atomic resolution 3D structure of the open hub to support this proposal. Parallel MD simulations show that such a structure would be created by a dimer loss from a closed 14-mer hub (Supplementary Fig. 6). The 3D-flex analysis and the MD demonstrate that once the dimer is removed, the hub will open further as more distant contacts are progressively weakened. An earlier simulation on a 10-mer open hub constructed in-silico from a crystal structure of a 12-mer γ hub suggested how this construct could similarly open further upon dimer loss[16], but experimental data were not available then for a detailed comparison of the dynamics. A comparison of the closed and open hub structures (Fig. 6) revealed an out-of-plane relaxation of the open hub. The possibility of such deformation was proposed previously for the α hub based on the spiral *S. rosetta* hub structure[18].

The stoichiometry in X-ray crystal structures reported for mammalian α and β hub polymorphs varies between 12 and 14 ADs. The AD stoichiometry in one crystal structure of the γ isoform differs from the stoichiometry reported in another single crystal structure of the δ isoform[10]. It is difficult to determine to what extent the variations are due to crystallization conditions. We, therefore, performed MD simulations to determine how the angular curvature of closed rings deviates from the intrinsic curvature of the tetramer, the first precursor with LC. We found the intrinsic LC curvature is intermediate between the curvature of the 12-mer and 14-mer hubs, consistent with their native ESI-MS 1:1 ratio for the human α hub[18]. The interfacial contact energetics of all near-atomic resolution hub structures were found to be indistinguishable between isoforms. This result predicts that comparable resolution structures from cryo-EM studies, when available, will show the coexistence of hub polymorphs for both γ and δ isoforms.

Based on our structures and the cumulative evidence in the literature, we can propose an integrated model for hub assembly/disassembly shown in Fig. 8. It postulates that hub polymorphism is an intrinsic biophysical property of all mammalian hubs with the dominant intrinsic symmetry set only by the free energy for hub closure (Δ*E*) and the concentration (*C*) of the available subunit pool. Interactions with other structural elements in the holoenzyme or extrinsic modulators are not considered. All reactions are reversible. Its three fundamental tenets are:

1. The assembly unit is the dimer with an intact VC[16] (Figs. 5 and 7 from this study).
2. The energy cost for ring closure depends on the strain imposed by the β-sheet deformation of the closed ring ADs[18]. Closure, an isomerization reaction, constraints flexure relative to the open precursors, the tetramer, and the cost will scale with the extent of the deformation from the open precursors. Closure will be stabilized by the formations of the two LCs. The tetramer precursor with both the LC and VC has intrinsic curvature intermediate between the 12-mer and 14-mer hubs. Larger or smaller hubs will have greater strain, hence energetic cost. The net energy for hub formation (Δ*E*) will decrease accordingly.
3. The increase in the size of open polymers depends on the dimer concentration (*C*). Mass spectrometry has shown that serial assembly of precursors to the open *S. rosetta* hub depends on dimer

concentration[18]. Dilution favors the disassembly of the rat β hub (This study (Fig. 5)) consistent with a similar mechanism.

In living cells, factors such as macromolecular crowding[29] and allosteric modulators[30], in addition to the KD, would become relevant. Nevertheless, the concentration dependence on the dimer pool is an order of magnitude higher in 14-mer relative to the 12-mer. This might explain why the subunit stoichiometry of Venus-tagged CaMKIIα holoenzyme measured in a diluted cell homogenate had approximately 2 fewer subunits than the holoenzyme stoichiometry measured in the intact cells that the homogenate was made from, while heterogeneous CaMKII stoichiometries were reported in live cell populations[31]. The stoichiometry reported by brightness analyses of plasmid-encoded α holoenzymes in homogenates ($n = 10$)[32] also agrees with the mean stoichiometries reported by TIRFM for our holoenzyme samples.

Intrinsic subunit polymorphism has consequences for earlier ideas proposed in the literature. The idea that subunit stoichiometry is fixed between isoforms with hetero oligomerization ratios used to optimize frequency decoding[9] is now unlikely. In addition, while the KD does affect holoenzyme stability, it does not switch the hub stoichiometry as speculated[14]. What, then, might be the function of intrinsic lability? The dynamic exchange will allow adjustment over minutes of hetero-oligomer αβ ratios by local synthesis upon synaptic stimulation[33,34] to influence dendritic spine cytoskeletal morphology and dynamics[35]; in line with the idea proposed long ago that "the synaptic localization of CaMKIIα activity is controlled by the relative expression of CaMKIIβ docking modules" in αβ hetero-oligomers[23]. Thus, while activation-triggered subunit exchange may turn out not to be important for structural long-term potentiation[36], intrinsic subunit exchange might be.

## Methods

### Plasmid construction

All bacterial protein expression plasmids were constructed in the pSMT3 vector backbone[13]. For 6His-SUMO-V-rCaMKIIβ, the open reading frame of the yellow fluorescent protein Venus was fused in-frame between the 6His-SUMO coding sequence of the SMT3 vector and the open reading frame of the rat CaMKIIβ[27] codon optimized for bacterial expression. The 6His-V15-rCaMKIIβ plasmid was created from 6His-SUMO-V-CaMKIIβ by removal of the SUMO tag and truncation of the V-rCaMKIIβ linker to 15 amino acids. The 6His-V15-mCaMKIIα holoenzyme constructs were created from 6His-SUMO-V-rCaMKIIβ by replacing V-rCaMKIIβ with V15-mCaMKIIα[37] and removal of the SUMO tag. The dimeric 6His-V15-mCaMKIIαF394A was created from 6His-V15-mCaMKIIα by site-directed mutagenesis. The 6His-V15-rCaMKIIβ and 6His-V15-mCaMKIIα hub constructs were created from the corresponding holoenzyme constructs by removal of the kinase domain. The monomeric 6His-V15-rCaMKIIβ was created by the removal of the association domains starting with residue 315. The dimeric βF458A, the constitutive phospho-mimic (T287D), and the dephosphorylated triple mutant mimic (T287A/T306A/T307A) were constructed from 6His-V15-rCaMKIIβ by site-directed mutagenesis. Primers and plasmid construction details are in Supplementary Table 1.

### Protein purification

Reagents were sourced from Sigma-Aldrich (Gillingham, Dorset, UK), except for mouse monoclonal GFP antibody no. 1814460 (Roche, Basel, Switzerland) and GFP protein, no. 8365-1 (Clontech (Mountain View, CA). The plasmids were co-transformed with a plasmid encoding protein phosphatase into BL21/Rosetta or BL21/C41 with double antibiotic (kanamycin/streptomycin) selection. Expression was monitored by Venus fluorescence in cell pellets from 20 ml Luria broth (LB) supplemented with antibiotics after IPTG (1 mM) induction. Colonies with the best yield were stored as glycerol (8%) stocks at −80 °C. The calcium calmodulin-dependent primary autophosphorylation of the native rat α and β V-CaMKII were checked with Abcam anti-CaMKIIα phospho-T286 antibody (ab124880) western blot on the Jess electrophoresis capillary system (ProteinSimple, San Jose, CA). The proteins were expressed in HEK cells in 12-well plates. The cells were harvested after 24 h. Western blots compared the autophosphorylation in $Ca^{2+}$ plus 1 mM ATP versus control buffers.

The proteins were harvested from 1 L LB bacterial cultures (BL21/Rosetta) after IPTG induction and 18 °C overnight incubation. All buffers were made in Milli-Q double-distilled water. The cell pellets were resuspended in low imidazole buffer A (25 mM Tris, 40 mM imidazole, 150 mM KCl, 0.2 mM TCEP, 10% glycerol, pH 8.5) supplemented with protease inhibitors (Sigma Fast, cocktail) and lysed with a cell disrupter (Constant Systems, Ltd, Daventry, Northants, UK). The lysate was clarified by centrifugation (40,000×$g$, 1 h, 6 °C), passage over a 5 ml nickel Hi-Trap column, and elution with high imidazole buffer B (Buffer A with 400 mM instead of 40 mM imidazole). The peak fractions were diluted 4x in buffer C (25 mM Tris, 25 mM imidazole, 100 mM KCl, 0.2 mM TCEP, 2% glycerol, pH 8.5), concentrated (Amicon. 300kD MW cutoff), then used immediately or flash-frozen and stored at −80 °C. The flash-frozen stocks were thawed on ice (4 °C) for EM and gel-filtration of the isolated hubs, as well as the TIRFM measurements. EM grid preparation for the holoenzyme samples used fresh, rather than flash-frozen preparations.

The gel-filtration buffer (50 mM Tris, 150 mM KCl, 0.2 mM TCEP, 2% glycerol, pH 8.0) was degassed and filtered (0.22 μm). The thawed stocks were applied without dilution to the SEC-650 gel-filtration columns. In contrast, they were diluted >1000× in AB⁻ buffer for the TIRFM experiments. The AB⁻ buffer (25 mM imidazole–HCl pH 7.5, 25 mM KCl, 1 mM EGTA, 4 mM $MgCl_2$. 2% glycerol) was degassed and supplemented with oxygen scavenger mix (0.2 mg/mL glucose oxidase, 0.5 mg/mL catalase, 3 mg/mL glucose, 0.5 mg/ml bovine serum albumin (BSA)). The BSA acted as a blocking agent. The gel-filtration and TIRFM experiments were conducted at room temperature.

### Gel Filtration

The samples (250 μL volume) were run on the EnRich SEC 650 Bio-Rad high-resolution size exclusion column (10 × 300 mm) equilibrated with gel-filtration buffer with a flow rate of 1 mL/min. The Bio-Rad size exclusion standards (Cat#151-1901) were used for calibration of the column. The Bio-Rad NGC chromatography system was equipped with multiple UV (280 nm)/visible (515 nm) wavelength detectors.

### Single-molecule TIRFM

The TIRFM experiments were part of a larger study to characterize the interactions of the rat CaMKII β isoform with calcium calmodulin that is reported elsewhere[38]. The basic design of the TIRFM workstation has been described[39]. The microscope flow cell, two coverslips held together by parallel strips of double-sided adhesive tape, was mounted on a 3-axis piezo stage (XYZ-SLC17:22 with MCS-3c, SmartAct, Oldenburg, Germany). The coverslip surface for total internal reflection of the 488 nm diode laser (Lighthub-6, Omicron, Rodgau-Dudenhofen, Germany) was sparsely populated with GFP antibody before adsorption was blocked by BSA. An aliquot of the V-CaMKII stock, freshly thawed at 4 °C from flash-frozen stocks, was diluted >$10^3$-fold with AB⁻ buffer was perfused into the flow cell. After incubation for 5 minutes, unbound V-CaMKII was washed out with AB⁺/O₂-scavenger mix[27]. The total experiment duration did not exceed 30 min. The 488 nm diode laser (Lighthub-6, Omicron, Rodgau-Dudenhofen, Germany) laser and camera (iXON-3, EMCCD, Andor, Belfast, Northern Ireland) were computer-controlled with a modified version of GMimPro[40].

The video records of the Venus fluorescence emission were acquired as described[38] and analyzed in ImageJ (version 1.53a) (Supplementary Fig. 7). Background intensity variation across the field of view was removed for each frame in the video stack using a rolling 100-pixel, ImageJ "Subtract background" function. Fluorescent spot locations were detected from the convolution of the averaged video stack with a $9 \times 9$ pixel$^2$ Laplacian of a Gaussian (LoG) matrix. Spots with intensity values in the top one percentile were selected for further analysis. Their locations were expanded to $5 \times 5$ pixel$^2$ regions, consistent with the expected Airy disc point spread function

**Table 1 | Cryo-EM data collection, refinement and validation statistics**

| | α hub 14-mer | α hub 12-mer | β hub 14-mer | β hub 12-mer | β holoenzyme 14-mer | β holoenzyme open 12-mer |
|---|---|---|---|---|---|---|
| *Data collection and processing* | | | | | | |
| Magnification | 165,000 | 165,000 | 105,000 | 105,000 | 105,000 | 105,000 |
| Voltage (kV) | 300 | 300 | 300 | 300 | 300 | 300 |
| Electron exposure (e⁻/Å²) | 60 | 60 | 60 | 60 | 60 | 60 |
| Defocus range (μm) | −0.6 to −1.5 | −0.6 to −1.5 | −0.6 to −1.5 | −0.6 to −1.5 | −0.6 to −1.5 | −0.6 to −1.5 |
| Pixel size (Å) | 0.74 | 0.74 | 0.86 | 0.86 | 0.86 | 0.86 |
| Symmetry imposed | D7 | D6 | D7 | D6 | D7 | C1 |
| Initial particle images (no.) | 2,244,235 | 2,024,671 | 2,437,000 | 1,882,694 | 2,074,285 | 2,074,285 |
| Final particle images (no.) | 205,902 | 204,506 | 126,970 | 185,091 | 138,904 | 332,360 |
| Map resolution (Å) | 2.6 | 2.7 | 2.6 | 2.6 | 3 | 3.5 |
| FSC threshold | 0.143 | 0.143 | 0.143 | 0.143 | 0.143 | 0.143 |
| Map resolution range (Å) | 1.9–3.0 | 1.8–3.3 | 1.9–3.0 | 1.9–3.1 | 2.1–3.5 | 2.1–7.9 |
| *Refinement* | | | | | | |
| Initial model used (PDB code) | Alphafold | Alphafold | Alphafold | Alphafold | Alphafold | Alphafold |
| Model resolution (Å) | 2.63 | 2.77 | 2.66 | 2.68 | 3.05 | 3.76 |
| FSC threshold | 0.5 | 0.5 | 0.5 | 0.5 | 0.5 | 0.5 |
| Map sharpening *B* factor (Å²) | −127.6 | −124.9 | −102.1 | −99.4 | −119.5 | −111.5 |
| Model composition | | | | | | |
| Non-hydrogen atoms | 15,218 | 13,044 | 15,106 | 12,948 | 15,106 | 12,948 |
| Protein residues | 1876 | 1608 | 1890 | 1620 | 1890 | 1620 |
| Ligands | | | | | | |
| *B* factors (Å²) | | | | | | |
| Protein | 33.78 | 84.98 | 20.35 | 29.18 | 40.27 | 48.6 |
| Ligand | | | | | | |
| R.m.s. deviations | | | | | | |
| Bond lengths (Å) | 0.006 | 0.005 | 0.007 | 0.005 | 0.005 | 0.011 |
| Bond angles (°) | 1.993 | 1.439 | 0.87 | 1.336 | 0.703 | 5.564 |
| *Validation* | | | | | | |
| MolProbity score | 0.88 | 1.29 | 1.21 | 1.03 | 2.01 | 1.48 |
| Clashscore | 1.13 | 2.49 | 3.6 | 2.43 | 5.18 | 5.68 |
| Poor rotamers (%) | 0 | 0 | 0 | 0.87 | 4.35 | 0.87 |
| Ramachandran plot | | | | | | |
| Favored (%) | 97.73 | 96.21 | 97.74 | 98.5 | 96.24 | 96.99 |
| Allowed (%) | 2.27 | 3.79 | 2.26 | 1.5 | 3.76 | 3.01 |
| Disallowed (%) | 0.00 | 0.00 | 0.00 | 0.00 | 0.00 | 0.00 |

(PSF). The regions were incorporated in a full frame Boolean mask to extract the intensity versus time record at 100 ms resolution for every selected spot.

Downstream analysis of the spot records with a 4-pass custom step finder algorithm identified photobleaching steps. The unitary Venus fluorophore intensity ($I_{UNI}$ ~ 175 counts/pixel) was determined in control experiments that used low densities of the monomeric Venus-tagged KD under identical imaging conditions. The $I_{UNI}$ intensity distribution guided step detection. The algorithm had a graphical interface with interactive "sliders" for manual optimization and superposition of the modeled step-wise intensity changes with the experimental record in real-time. Once the slider parameters had been optimized with a small subset, they were fixed for analysis of records in the entire experimental dataset. The average photobleaching step amplitudes for the dimer assemblies were comparable to the unitary Venus fluorescence as expected. Single photobleaching steps were easily identified at later times in the holoenzyme spot records after subtraction of the autofluorescence decay, as explained (Supplementary Fig. 7). The subunit number of the V-CaMKII contained within each spot was obtained by the number of detected steps plus the number obtained by division of the residual spot intensity with the average step amplitude.

**Cryo-EM sample preparation and data collection**

Purified samples with a concentration of 20 mg/mL were used for cryo-EM grid preparation. 3 μL of the sample was applied to glow-discharged Quantifoil R2/1 Cu200 grids, then blotted with Whatman filter papers for 3 s before plunge-frozen in liquid ethane using an FEI Vitrobot Mark IV (Thermo Fisher Scientific) at 4 °C and 100% humidity. Cryo-EM data were collected with Thermo Fisher Scientific Titan-Krios cryo-electron microscopes operating at 300 keV. CaMKIIα hub data were collected with a Falcon 4 camera and Selectris energy filter set to 10 eV. CaMKIIβ hub and CaMKIIβ holoenzyme data were collected with a Gatan K3 camera and Bioquantum energy filter set to 20 eV. Automated data collection was performed using the Thermo Fisher Scientific EPU software with parameters reported in Table 1.

## Image processing and model refinement

The complete data processing workflow is reported in Supplementary Figs. 8–10. Data pre-processing (motion correction and CTF correction) was carried out in CryoSPARC live[41]. Micrographs were curated by total motion, CTF fit, and relative ice thickness. The processing strategy is similar for the three samples. The first round of processing involved template picking in CryoSPARC followed by iterative rounds of 2D classifications, ab-initio reconstructions, and heterogeneous refinements to filter out unwanted particles. The particles retained were re-extracted and refined using non-uniform refinement in CryoSPARC to confirm their high quality. These particles were then used to train Topaz Picker[42]. For the CaMKIIβ holoenzyme sample, the preferred orientation was identified (streaking feature in the reconstructed map). As a result, an additional Topaz trained using side view particles was used to enrich the missing views. The particles picked by Topaz went through the same pruning process as before to filter out junk particles. The final three-dimensional reconstructions were done using non-uniform refinement with symmetry imposed in CryoSPARC. The reported resolutions were based on the 0.143 Fourier Shell Correlation (FSC)[43]. 3DFlex analysis[22] for open ring structure was done within CryoS-PARC. Initial atomic models were predicted by AlphaFold2[44] using the monomer AD sequence, then symmetrized to generate the hub models. They were rigid-body docked into the cryo-EM maps and refined using Refmac5[45] in CCP-EM suite[46] or Phenix[47]. Atomic models were improved by ISOLDE[48]. The closed 14-mer and open 12-mer β hub models were related within UCSF ChimeraX[49] by the Morph utility[50] to map hinge motions and the Matchmaker utility[51] for superimposition of the extracted ADs. All atomic models were validated by MolProbity[52] and Q-score[20]. The RMSD values for comparing α hub and β hub models were obtained using Matchmaker with all default settings. They were calculated on 129 pruned Cα pairs (out of the total 133 pairs). All structure figures were prepared with UCSF ChimeraX[49].

## Molecular dynamics (MD)

The simulations were conducted with NAMD and the CHARMM27 force field and TIP3 water model[53] on the NIH HPC Biowulf supercomputer. The atomic structures were neutralized with 150 mM sodium chloride at pH 7.0 in a water box with periodic boundary conditions. Replicate simulations followed the canonical NVT protocol[54] at 310 K temperature, 1 atm pressure, The temperature was maintained by a Langevin piston with a damping constant of 1 K/ps. The particle mesh Ewald (PME) grid spacing was 1 Å with a summation cutoff of 12 Å. Initial minimization (400 frames) and equilibrium (40 ps) at 2 fs/frame, were followed by 0.4 ns pre-production runs with NAMD/2.14-verbs on the CPU multimode partition. Production runs (20 – 200 ns) were conducted with GPU NAMD version 3.0[53]. The MDtraj mdconvert utility[55] converted between trajectory formats. Trajectories were edited to remove the water before analysis with GROMACS 2020 gmx rms, gmx rmsf, and gmx sgangle functions[56]. The α hub simulations were sampled at 1 fs/frame for Q-score comparison with the RMSF. The number and duration of replicates for each assembly, together with the assembly and water box size, are listed in Supplementary Table 2a. The Table includes an illustrative trace of an initial energy minimization run for the open β hub. Short descriptions of previously published simulations that motivated the present study are given in Supplementary Table 2b.

The MD was supplemented with ePISA[25] comparison of our static hub structures with those obtained by X-ray crystallography. The structure sequences were aligned with MUSCLE[57]. Plots were prepared with Xmgrace (https://plasma-gate.weizmann.ac.il/Grace) and Sigmaplot version 12.0 (Inpixon, Palo Alto, CA).

## Statistics and reproducibility

For the cryo-EM experiments, four grids were typically frozen and screened per experiment. Particle sample sizes at various stages of the reconstruction of the 3D maps are given in Supplementary Figs. 8–10. For the single particle LM distributions shown (Fig. 5), the F-test was used to catalogue the differences between them. The number of experiments, records per experiment and spot population size are given in the "Methods" section or the figure legend. For the MD simulations, the number of replicates for each assembly is given in Supplementary Table 2. The Pearson correlations between the RMSF and Q-scores were averaged over replicates. The angle measurements for the tetramer assemblies were also averaged, with standard errors (SEs) shown for the individual distributions (Fig. 4) to distinguish between them. For the angle-time trajectories of the open structure replicates for which convergence is not expected, the independent trajectories from three replicates are shown (Supplementary Fig. 6).

## Reporting summary

Further information on research design is available in the Nature Portfolio Reporting Summary linked to this article.

## Data availability

Cryo-EM density maps have been deposited to the Electron Microscopy Data Bank (https://www.ebi.ac.uk). The depositions include the map of the 12-mer close hub (EMD-43385). Atomic coordinates of the models have been deposited to the Protein Data Bank (https://www.rscb.org). α-14-mer (40873 EMD, 8SYG PDB), α-12-mer (40955 EMD, 8T15 PDB), β-14-mer (40956 EMD, 8T17 PDB), β-12-mer (40957 EMD, 8T18 PDB), β-14-mer holoenzyme (41070 EMD, 8T6K PDB), β dodecamer open ring (41077 EMD, 8T6Q PDB). MD simulations, shortened to comply with the repository size limit, have been uploaded to Figshare (https://figshare.com/s/c59a09308aeebbeda44b). Example video records and text files for the photobleaching data and simulations have been deposited in GitHub (https://github.com/GoToJustin/Khan_et_al_2023).

## Code availability

The code for the single-molecule photobleaching analysis can be downloaded from the same GitHub site.

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

## Acknowledgements

This study was funded, in part, by the intramural program of the National Institutes of Health (NIH), National Institute on Alcohol Abuse and Alcoholism (NIAAA) (NIH 1-Z01 AA000452 to S.S.V.); NIH grants

(P41GM103832 and R01GM079429 to W.C.), and the Francis Crick Institute (J.E.M); which receives core funding from CRUK (CC119); MRC (CC119); Wellcome Trust (CC119). We thank Dr Kirk HInes (NIH/NIAAA) for assistance with the gel filtration experiments. The cryo-EM data collection was performed at the Stanford-SLAC Cryo-EM Center (S$^2$C$^2$), which was supported by the NIH Common Fund Transformative High-Resolution Cryo-Electron Microscopy program (U24 GM129541). The MD simulations utilized the NIH HPC Biowulf computational cluster (http://hpc.nih.gov). NAMD was developed by the Theoretical and Computational Biophysics Group in the Beckman Institute for Advanced Science and Technology at the University of Illinois at Urbana-Champaign.

## Author contributions

C.-T.C. conceived and executed the cryo-EM experiments, processed the cryo-EM data, and wrote the manuscript. H.P. made all plasmid constructs, advised on protein expressions, and contributed to manuscript drafts. S.S.V. assisted with the design and analysis of the single-molecule fluorescence experiments and edited the manuscript for submission. J.E.M. wrote the code for the analysis of the single-molecule fluorescence data, assisted with experimental design and interpretation, and edited the manuscript for submission. W.C. advised on all aspects of the cryo-EM, edited the manuscript, and verified that the figures and conclusions accurately reflected the original cryo-EM image datasets and arranged for their preservation. S.K. performed MD simulations, executed and analyzed the single-molecule fluorescence experiments, wrote the manuscript, and was responsible for the overall design of the project and data availability.

## Funding

## Competing interests

The authors declare no competing interests.
