## [Peer Review File · Communications Biology]

Reviewers' comments:

Reviewer #1 (Remarks to the Author):

Calcium calmodulin-dependent kinase II (CaMKII) forms higher order dihedral assemblies through its C-terminal association domain, with the N-terminal kinase domains splayed outwards on a flexible linker. While there have been several crystal structures of both domains, the dependence of X-ray crystallography on homogenous populations for crystallisation has obscured the oligomeric heterogeneity of CaMKII isoforms that may be relevant for their biological functions. Here the authors take advantage of cryo-electron microscopy (cryo-EM) to capture multiple oligomeric states of two isoforms of CaMKII, producing the first high-resolution cryo-EM structures for this complex (2.6-3.0 Å vs. previous best at 4.8 Å). Importantly, they solved a structure of a partially dissociated assembly not seen before by X-ray crystallography structures. These structures are supplemented by molecular dynamic simulations and total internal reflection fluorescence microscopy experiments to study the association and dissociation dynamics of the complex. Together it points to a model of oligomeric assembly beginning with a dimer of CaMKII which progressively forms a ring via flexible lateral interfaces, and demonstrates the value of using cryo-EM to study the heterogeneity of these assemblies.

I have some issues with the figures and text as detailed below.

Comments:

1. There is insufficient background on previous structures in the introduction to provide context to the work. Other structures are briefly mentioned as 'hubs trapped in different oligomeric states depending on isoform, engineered residue substitutions, or presence of the KD' but there is no detail on what oligomeric states have been previously observed i.e. are the proportions of 12-mer vs. 14-mer in this study consistent with previous studies? I recommend expanding this as well as including an extended data figure or table with an overview of existing structures alongside the new structures.
2. Due to the presence of a long linker, the kinase domain is flexible and not observed in the cryo-EM reconstructions of the holoenzymes but the presence of this domain is not indicated in any of the figures e.g. as a cartoon. A diagram of the whole assembly would help explain the assembly especially the autoinhibition by the R alpha-helix which is central to the authors proposed mechanism.
3. The use of dihedral symmetry (e.g. D6, D7) to describe the hub/assembly is unusual for CaMKII structures, with most papers describing them as 12-mer or 'dodecameric' assemblies. This is because a small break in symmetry, such as 'breathing' or 'flexing' of the assembly, would not affect the oligomerisation state (12-mer) and function, even if D6 symmetry is no longer strictly valid.
4. Similarly, the use of the singular 'vertical contact' to describe the interface is unusual c.f. terms like contact area, surface interaction, contact residues, interface, which are more prevalent in the literature.
5. In the discussion, the variability in oligomerisation is proposed to be due to the energetic cost of ring closure and expression level. However, expression level is only one factor dictating protein

concentration in cells e.g. sequestration, recruitment, transport, degradation etc. I think it would be more appropriate to describe a concentration-dependence for the oligomerisation, rather than expression levels.

6. Several of the structures have missing density for loops and N- and C-terminal ends, yet these residues are present in the model for PDB/EMDB validation. This is highlighted in Section 9 of the validation reports. While it is appropriate to show in figures an overlay of an AlphaFold model or known structure and a map with missing density relative to what is expected e.g. Fig 1c, I seriously question the deposition of a model containing residues with insufficient density to model the backbone. This is especially important for the open-ring structure, which has missing density for some domains. In some cases it may be that the 'recommended contour level' of the maps is too high (see Sections 7.2 and 9.4 of Validation reports).

7. The PDB Validation reports also contain discrepancies between the modelled and reference sequence which seems to be GFP, rather than CaMKII.

8. There is a lack of biochemical data shown for the purification of the proteins, such as an SDS-PAGE gel to validate the purity of the samples used for cryo-EM and microscopy experiments.

9. There are missing symbols throughout the text, possibly due to poor encoding or alternative fonts in the PDF.

10. Lines 69-70: "Our cryo-EM structures of the isolated β hubs complement those studies." It should be clearly delineated in the text that these structures are of Rat CaMKII.

11. Lines 70-72: Different font, missing symbol.

12. Line 96: 'D' should be explained as dihedral symmetry at least once in the text

13. Line 119: Missing reference to Extended Data Fig 1e where a comprehensive comparison can be found.

14. Line 120: 'The LC occurs between α -helices $\alpha 2$ and $\alpha 3$...' – There should be at least one figure with the alpha helices and beta-strands numbered so this description makes sense.

15. Lines 127-128: 'The F-Q interaction is the key contributor to LC stabilization for other hub structures as the hydrogen bond between F-Q was reported by ePISA for all structures.' – this analysis/data should be included in the supplementary

16. Lines 136-137: 'I434 in α is equivalent to L498 in β . (Extended Data Fig. 4)' specify the figure panel and include Fig 1e which has I434.

17. Lines 134-135,207-208: Detail the high resolution limit for which the correlation was calculated.

18. Lines 196-199: Here R is referred to as a peptide, implying a separate molecule rather than the connected alpha-helix in Line 61.

19. Line 204: This 'low-resolution dodecamer closed ring reconstruction' of the holoenzyme is not displayed in Ext. Data Fig 5 and I also cannot find it in the workflow diagrams. The specific resolution of this dodecamer should be noted, along with the relative proportions of these three assemblies in the dataset.

20. Figure 5: y-axis labels are cropped in panels a and e.

21. Figure 6: The legend describes four possible scenarios, but the figure is a composite diagram. A four-panel figure describing the different scenarios or clearer labelling is needed. Several elements of the figure are not explained in the legend e.g. 'C', ' ΔE '.

22. Line 433: The source of the D8 map depicted is not given.

23. Line 571: For the alphafold initial models, were these made using the whole sequence of the holoenzyme and was this used to generate a monomer or oligomer initial model?

24. Ext. Data Fig 1 – The sequence alignment image is low resolution/distorted.
25. Ext. Data Fig 2 + 3 – No scale bar for panel b+c. Panel d + g local resolution figures lack labelling of Resolution (Å) for color key. Panel e + f missing 0.143 label, and axes labels.
26. Ext. Data Fig 7-9 (Cryo-EM Workflow Diagrams) – Although motion correction and CTF-estimation are mentioned in the methods they are not listed here, and there is no indication of the starting number of movies and how many were retained after micrograph curation. Also, when the dataset is partially processed and then a 2D class/3D map is used to re-pick the micrographs, the workflow should link back to the micrographs i.e. fork from the top of the diagram, rather than continuing in a linear arrangement.
27. Ext. Data Fig 9: What looks like a 2D-class of a side view is used for Topaz picking and combined with a separate picking job, but this is not explained in the text or legend. It's not clear why this was done here and not in the other workflows.
28. Supp. Videos – Indicate which video represents latent coordinate 1 or latent coordinate 2 as shown in Figure 3.

Reviewer #2 (Remarks to the Author):

This interesting paper takes a three-pronged approach to examine the oligomerization state/stability of CaMKII holoenzymes. The three approaches are single-particle cryo-EM, molecular dynamics simulations and single-molecule fluorescence microscopy. The overall conclusion is a proposed assembly model where CaMKII goes through a dimer->tetramer intermediate on the path to a dodecameric or tetradecameric assembly. The cryo-EM studies appear well-done and show results consistent with other crystal structures and negative-stain single-particle reconstructions. Both dodecameric and tetradecameric assemblies of alpha and beta hubs (constructs lacking the kinase domain) and a full-length betaCaMKII were assessed. Interestingly, a form of the full-length betaCaMKII with an open hub structure was a large proportion of the particles and along with the mixtures of 12- or 14-mer hubs indicates an instability in the hub domain. The description of vertical and lateral domain interfaces is not new given there are higher resolution crystal structures available to draw these conclusions but is worthy of commenting on in the context of this paper. The resolution of the reconstructions is impressive and pushing the SPA with cryo-preserved specimens is an important advance, although there are some issues to address. The molecular dynamics simulation is a nice compliment to the structural work, but don't seem they are dependent on the new data presented in this manuscript with one exception. The simulations of the open structure for betaCaMKII holoenzymes seems interesting, but the conclusion on the spreading of the instability seems speculative. The single-molecule fluorescence studies are also interesting and further support that there is instability in CaMKII. The results are complex and appear to depend heavily on model-based data analysis. While intermediates are clearly identified and mutations and nucleotide appear to alter the stability, the exact counting of molecules beyond a monomer/dimer seem difficult to accurately deduce. A general overall comment is that while the three approaches are complimentary, the manuscript, particularly the results, reads like three separate papers. There is also insufficient presentation/integration of a significant body of literature to judge the novelty/impact of the present manuscript.

Critique:

Major

1. The primary data for the cryo-EM figures that leads to the SPA analysis is problematic. The images appear to be a tangled mass of protein and it's not clear what represents single particles on which all subsequent analysis is based. Better images of the primary data processed are needed with clearly identified single particles.
2. Also, for the SPA, It is not clearly stated if each of the reconstructions comes from a single preparation of each construct, or if attempts were made to analyze multiple preparations of each purified protein. One important issue here is how consistent the distribution of D6 and D7 particles is across different preparations. Perhaps most importantly is whether the betaCaMKII holoenzymes where the open hub was the dominant (75%) form reproducible. Related is whether these D6 and D7 or open structures reported are the result of bacterial expression/protein purification. This does not seem considered anywhere in the manuscript. Compounding the concern is lack of indication of how reproducible the findings are between preparations.
3. Is there a reason that the assessment of the dodecameric CaMKIIbeta preparations was abandoned (page 10 lines 204-205)? This data should be presented even if not achieving the resolution of the other reconstructions.
4. Clearer statements need made to the claims of resolution for the CaMKIIbeta holoenzyme samples. It is not possible that the claimed resolutions are reflective of the entire holoenzyme as the KDs and then Venus tags were not resolved at all. The writing needs clarified here that the resolutions are specifically related to the hub domains and should identify exactly which residues the resolution claim applies to.
5. There is also no indication that the enzymatic activity of these preparations was assessed and whether the activity is consistent across different preparations. Some validation of the enzymatic activity of these preparations (when the KD is present) is essential.
6. This reviewer does not have the expertise to comment on the technical merits and findings from the MD simulations. The most intriguing observation appears to be the findings in Figure 4 where there is spreading of the instability shown by an increased B-factor. How consistent is this finding? Further, it does not appear the system has reached stability after 100 ns, how does this influence the interpretation of the results?
7. The description of the data concerning the single-molecule fluorescence is difficult to follow. There seems no need to summarize the work before presenting each of the analysis.
8. There is a similar challenge for the fluorescence studies as noted for the SPA analyses. How were the preparations characterized for their enzymatic activity and how many separate preparations were assessed to evaluate the reproducibility of the findings?
9. The experiment mostly directly assessing the stability of the oligomeric state is the aging experiment described in section C. However, the design is unclear. Was the preparation applied to the chip as single holoenzymes (or dissociated subunits) and then imaged at two different times, or diluted, aged, and then applied to the chips? Related, if the instability is as great as suggested why aren't the single bound molecules falling apart during the imaging protocol. Or are they and that explains some of the complex disappearance of fluorescent signals?
10. The discussion of energies and hub stability in the discussion (lines 390 to 407) are convoluted

and difficult to follow. If there are energetic penalties going from D7 to D6 then why would one ever see any significant fraction of 10-mers (D5?) noted in reference 18. Also in the middle of the same paragraph is a passing note about proteolysis that might influence the analyses. This might be true, but then an assessment of the state of the molecules under analysis using orthogonal techniques seems warranted.

11. Why do the authors dismiss specific residue differences in the hub domains leading to differences in the final oligomeric structures? Lines 384-386? There is a passing comment about crystallization conditions being an issue that makes little sense.

12. The thought expressed in lines 386-389 is additionally unclear.

13. Line 418 suggests that the present study “supplants” earlier ideas. I do not understand this claim. The study is based on, and modestly extends many earlier studies. The other points in the conclusionary paragraph seem a collection of not well-developed thoughts about LTP, local protein synthesis, role of the R peptide...etc. Overall the Discussion needs a major rework to improve the integration of the findings in this paper from earlier literature.

Minor

1. Page 10, line 196 – the mutations do not eliminate endogenous phosphorylation. The eliminate phosphorylation of specifically those sites.

2. Page 18-19 lines 383-384 the finding of the gamma and delta isoforms are reversed. In the Rellos et al. paper, gamma is D6 and delta is D7.

Reviewer #3 (Remarks to the Author):

The authors investigated the structure and stoichiometry of CaMKII beta hub domain and holoenzyme using cryoEM, MD simulations, and single molecule fluorescence microscopy. CaMKIIalpha hub structures (fused to Venus) are reported using cryoEM, and they show both 12mer and 14mer stoichiometries. They simulate 14mer CaMKII alpha structure to compare the two interfaces that hold the hub together. CaMKII beta holoenzyme structures (fused to Venus) are reported using cryoEM, and interestingly they observe open ringed structures, which are further analyzed using MD. CaMKII beta holoenzyme fused to Venus is analyzed in bulk using single molecule TIRF paired with photobleaching. The most interesting and novel result is the open-ringed structure, but more details need to be provided about the sample itself as well as the data analysis. It also would have been nice to show this in a tag-less construct. The other results muddy this potentially exciting finding – especially the smTIRF, which seems out of place.

Main concerns

1. The authors use a wide repertoire of techniques to characterize CaMKII. However, as presented, the results do not flow well. The single molecule TIRF experiment is interesting on its own but it is unclear how it's related to the cryoEM results. The dimer mutations come out of nowhere, as does monitoring activation (by adding ATP).

The writing is also disjointed and incorrect at times. For example in the abstract they write “hub

subunit stoichiometry is important for kinase activity” – but it’s unclear what they mean by this. The kinase is activated well even when the hub is not present. I also find the title misleading as no energetics are performed in this paper.

2. The model presented in Figure 6 is problematic for several reasons. One is that the comparisons being made in the manuscript are of different variants and different fusion constructs. For instance, hub versus holoenzyme, and both are fused to a large fluorescent protein which may impact population size. Second, cryoEM is a fantastic tool for generating structures, but is not always representative of the populations in solution for technical reasons. Thus it is inappropriate to convert these populations to an energetic argument. As a side note, showing a structure of D8 here is misleading as that was not solved in this manuscript.

3. It’s quite surprising that so many particles of CaMKII beta holoenzyme are in an open conformation. Why are the authors adding 4 mM Mg²⁺ to these samples? Since this result is unexpected, the authors should show some validation that their sample is behaving well in solution under these conditions before adding to the grid. (Mass photometry, SEC-MALS, or something similar). It would be nice to show raw images in the supplement as well.

4. In the smTIRF experiment, the authors need to show representative images of their single molecules. They also need to clarify the conditions used in all these samples, as the buffer components are not explicitly stated everywhere (which of them have ATP/Mg, etc). The starting intensity of the dimer is ~300, whereas the starting intensity of holoenzyme is >4000. Based on the dimer intensity, this would indicate that each venus is ~150 and a 12mer should be 1800, and a 14mer should be 2100. The authors need to explain why the holoenzyme intensity is so much greater.

Minor points

- the data shown in Figure 2 is not so novel – this has been pointed out many times before in structures and MD
- different fonts and font sizes are used
- in Fig 4C, one subunit is colored white to indicate an open ring? Is this on purpose?

01/16/'24

Responses to Reviewers' Comments

***Reviewer Comments (Calibri 12 font)**
Responses (Georgia 12 font)

Reviewer #1 (Remarks to the Author):

Calcium calmodulin-dependent kinase II (CaMKII) forms higher order dihedral assemblies through its C-terminal association domain, with the N-terminal kinase domains splayed outwards on a flexible linker. While there have been several crystal structures of both domains, the dependence of X-ray crystallography on homogenous populations for crystallization has obscured the oligomeric heterogeneity of CaMKII isoforms that may be relevant for their biological functions. Here the authors take advantage of cryo-electron microscopy (cryo-EM) to capture multiple oligomeric states of two isoforms of CaMKII, producing the first high-resolution cryo-EM structures for this complex (2.6-3.0 Å vs. previous best at 4.8 Å). Importantly, they solved a structure of a partially dissociated assembly not seen before by X-ray crystallography structures. These structures are supplemented by molecular dynamic simulations and total internal reflection fluorescence microscopy experiments to study the association and dissociation dynamics of the complex. Together it points to a model of oligomeric assembly beginning with a dimer of CaMKII which progressively forms a ring via flexible lateral interfaces and demonstrates the value of using cryo-EM to study the heterogeneity of these assemblies.

We thank the reviewer for the positive comments.

I have some issues with the figures and text as detailed below.

Comments:

1. There is insufficient background on previous structures in the introduction to provide context to the work. Other structures are briefly mentioned as 'hubs trapped in different oligomeric states depending on isoform, engineered residue substitutions, or presence of the KD' but there is no detail on what oligomeric states have been previously observed i.e. are the proportions of 12-mer vs. 14-mer in this study consistent with previous studies? I recommend

expanding this as well as including an extended data figure or table with an overview of existing structures alongside the new structures.

The details, including oligomeric states, for the structures noted in the statement in question (Introduction 3rd paragraph) are given in Expanded Data Figure 4e. These X-ray structures do not inform on the relative proportion of the polymorphs in solution. This information is provided by mass spectrometry as now stated near the end of this paragraph.

2. Due to the presence of a long linker, the kinase domain is flexible and not observed in the cryo-EM reconstructions of the holoenzymes, but the presence of this domain is not indicated in any of the figures e.g. as a cartoon. A diagram of the whole assembly would help explain the assembly, especially the autoinhibition by the R alpha-helix which is central to the authors proposed mechanism.

A schematic of the whole assembly is now shown in the Introduction (Fig. 1). Fig. 1 (now Fig. 2) and Fig. 3 (now Fig. 5) clarify that only the AD hubs are visualized and analyzed. The experiment that considers the role of the R alpha-helix has been moved to another manuscript dedicated to R interactions with calcium-calmodulin (Khan et al., 2023). We agree with the concern of Reviewer-3 that this experiment distracted from the essential focus of the study, namely the intrinsic polymorphism.

3. The use of dihedral symmetry (e.g., D6, D7) to describe the hub/assembly is unusual for CaMKII structures, with most papers describing them as 12-mer or 'dodecameric' assemblies. This is because a small break in symmetry, such as 'breathing' or 'flexing' of the assembly, would not affect the oligomerization state (12-mer) and function, even if D6 symmetry is no longer strictly valid.

Agreed. D6 and D7 have been replaced with 12-mer and 14-mer respectively. These abbreviations are defined on first use (Results 1st paragraph).

4. Similarly, the use of the singular 'vertical contact' to describe the interface is unusual c.f. terms like contact area, surface interaction, contact residues, interface, which are more prevalent in the literature.

Point taken. "Vertical contact" and "lateral contact" were abbreviations to save space. Instead, LC and VC are now the acronyms for the lateral contact interface and the vertical contact interface respectively. They are defined on first use and serve the same purpose as the former terms.

5. In the discussion, the variability in oligomerization is proposed to be due to the energetic cost of ring closure and expression level. However, expression level is only one factor dictating protein concentration in cells e.g., sequestration, recruitment, transport, degradation etc. I think it would be more appropriate to describe a concentration-dependence for the oligomerization, rather than expression levels.

Thank you. Concentration is indeed the relevant variable. We specify that the polymorph ratio depends on the concentration in the revised Discussion.

6. Several of the structures have missing density for loops and N- and C-terminal ends, yet these residues are present in the model for PDB/EMDB validation. This is highlighted in Section 9 of the validation reports. While it is appropriate to show in figures an overlay of an AlphaFold model or known structure and a map with missing density relative to what is expected e.g. Fig 1c, I seriously question the deposition of a model containing residues with insufficient density to model the backbone. This is especially important for the open-ring structure, which has missing density for some domains. In some cases, it may be that the 'recommended contour level' of the maps is too high (see Sections 7.2 and 9.4 of Validation reports).

We have updated our PDBs to include B-factors, which were derived from Q-scores (Pintilie et al., 2020). The Q-scores have become standardized metrics to indicate atom resolvability now adopted in PDB reports. This will make it clear that some regions have lower resolution (i.e. lower Q-score and thus poorer resolvability). This is particularly true of loop regions that have greater flexibility (Fig. 2c). In addition, the 'recommended contour level' in the original validation report for the open-ring structure (0.127) was too high, resulting in a lower atom-inclusion level of 55%. These have been fixed in the updated validation reports with a contour level of 0.06, resulting in an atom inclusion level of 82%.

Furthermore, we improved our open-ring structure by refining the model against the unsharpened map. The original deposited model was refined against the sharpened map. The opening region, with lower resolution, was oversharpened in this map. As a result, the density becomes fragmented and difficult to interpret. In contrast, the unsharpened map shows clear and connected densities that are easier for secondary structure identification. Our updated model refined against the unsharpened map is more accurate which contributed to the improved atom inclusion level of 82%.

7. The PDB Validation reports also contain discrepancies between the modelled and reference sequence which seems to be GFP, rather than CaMKII.

Thank you. The reference sequences in the original PDB validation reports were indeed incorrect. We have corrected them. The discrepancies reported in the new validation reports are expected as they correspond to differences between enhancing mutation residue positions in Venus, an engineered GFP, with the native GFP sequence used for the default sequence alignment.

8. There is a lack of biochemical data shown for the purification of the proteins, such as an SDS-PAGE gel to validate the purity of the samples used for cryo-EM and microscopy experiments.

Analytical gel filtration data with superimposed protein and Venus absorption profiles are now presented in Extended Data Figs. 4g and 5b to document the purification. In addition, the gel filtration supports the TIRFM evidence for holoenzyme disassembly at room temperature.

9. There are missing symbols throughout the text, possibly due to poor encoding or alternative fonts in the PDF.

These errors in the initial submission were produced during the upload onto the Journal submission site. We apologize for not having proofread the uploaded versions more carefully. We have done so now.

10. Lines 69-70: "Our cryo-EM structures of the isolated β hubs complement those studies." It should be clearly delineated in the text that these structures are of Rat CaMKII.

Done. Introduction 4th paragraph.

11. Lines 70-72: Different font, missing symbol.

Corrected

12. Line 96: 'D' should be explained as dihedral symmetry at least once in the text

D6 and D7 have now been replaced with 12-mer and 14-mer as stated in the response to point-3 above.

13. Line 119: Missing reference to Extended Data Fig 1e where a comprehensive comparison can be found.

The Extended Data Fig. 4e is now cited in the revised statement (Results 5th paragraph last line).

14. Line 120: 'The LC occurs between α -helices $\alpha 2$ and $\alpha 3$...' – There should be at least one figure with the alpha helices and beta-strands numbered so this description makes sense.

The secondary structure elements, actually $\alpha 3$ and $\alpha 4$, are now labeled in Fig. 2c, numbered according to their position in the 5' -> 3' sequence.

15. Lines 127-128: 'The F-Q interaction is the key contributor to LC stabilization for other hub structures as the hydrogen bond between F-Q was reported by ePISA for all structures.' – this analysis/data should be included in the supplementary

This analysis is now reported in Extended data Figure 4f.

16. Lines 136-137: 'I434 in α is equivalent to L498 in β . (Extended Data Fig. 4)' specify the figure panel and include Fig 1e which has I434.

The requested changes have been made to the statement (now Results 4th paragraph last line (173-174)).

17. Lines 134-135,207-208: Detail the high-resolution limit for which the correlation was calculated.

We have replaced the map correlation coefficient with model RMSD values for a more rigorous structural comparison (Results Section 1, line 137, and Section 4, line 313). The alpha 14-mer hub and beta 14-mer hub have an RMSD of 0.3 Å. The alpha 12-mer hub and beta 12-mer hub have an RMSD of 0.35 Å. The beta holoenzyme 14-mer and isolated 14-mer hub have an RMSD of 0.52 Å.

18. Lines 196-199: Here R is referred to as a peptide, implying a separate molecule rather than the connected alpha-helix in Line 61

The lines refer to published work from another group (Karandur et al., 2020), not the current study. In that work, peptides derived from the R segment were added to isolated hubs with hub disassembly into smaller oligomers monitored by mass spectrometry.

19. Line 204: This 'low-resolution dodecamer closed ring reconstruction' of the holoenzyme is not displayed in Ext. Data Fig 5 and I also cannot find it in the workflow diagrams. The specific

resolution of this dodecamer should be noted, along with the relative proportions of these three assemblies in the dataset.

Agreed. We now note the resolution for the closed 12-mer (8.4 Å (lines 306-307)). We have updated the presentation of the maps from the holoenzyme sample (Figure 3 (now Figure 5)) and the image processing workflow (Extended Data Fig. 10) to include the closed 12-mer reconstruction. The workflow diagram lists the number of particles used at each processing step. The 2 million or so particles picked by Topaz were processed to obtain the final 3D reconstructions for the three assemblies; 14-mer, 12-mer and open 12-mer. From the heterogenous refinement result, the particle numbers were 427,485 (14-mer), 538,396 (12-mer) and 775,486 (open 12-mer). The rest were grouped in an unassigned 3D class. The number of particles used for the final maps was 2-3-fold lower than number from the heterogenous refinement result. If one assumes that the particles picked by Topaz are representative of the entire sample, the relative proportions are 21%, 26% and 37% respectively. However, losses due to denaturation at the air-water interface, poor ice quality and particle overlap complicate the extrapolation from the picked particles to the actual populations in the corresponding sample.

20. Figure 5: y-axis labels are cropped in panels “a” and “e”.

Fig. 5 (now Fig. 4) and the section with which it was associated has been substantially reduced. Experiments and simulations not directly related to hub disassembly have been moved to a separate manuscript. We have checked that the axes labels are preserved in the revised Fig. 4 and the associated Extended Data Figs. 5c-e.

21. Figure 6: The legend describes four possible scenarios, but the figure is a composite diagram. A four-panel figure describing the different scenarios or clearer labelling is needed. Several elements of the figure are not explained in the legend e.g., ‘C’, ‘ ΔE ’.

The four different scenarios (or outcomes) are all due to the balance between energetic cost (ΔE) for ring formation and the concentration of the dimer building block (C). The connectivity will be lost if the Figure is partitioned into panels. We have opted to keep the Figure intact but have expanded the legend to define all the Figure labels/maps and relate it more closely to the associated text.

22. Line 433: The source of the D8 map depicted is not given.

Thank you. The omission has been corrected (Figure 8 legend).

23. Line 571: For the AlphaFold initial models, were these made using the whole sequence of the holoenzyme and was this used to generate a monomer or oligomer initial model?

The initial AlphaFold2 models were made using the sequence of the association domain. The single domain models were then symmetrized to generate the hub models.

24. Ext. Data Fig 1 – The sequence alignment image is low resolution/distorted.

The sequence alignment image has been replaced with an enlarged, high-resolution image in the revised Figure (now Extended Data Fig. 4).

25. Ext. Data Fig 2 + 3 – No scale bar for panel b+c. Panel d + g local resolution figures lack labelling of Resolution (Å) for color key. Panel e + f missing 0.143 label, and axes labels.

Thank you. All omissions have now been included in the revised Figures (Extended Data Figs. 2,3, and 6).

26. Ext. Data Fig 7-9 (Cryo-EM Workflow Diagrams) – Although motion correction and CTF-estimation are mentioned in the methods they are not listed here, and there is no indication of the starting number of movies and how many were retained after micrograph curation. Also, when the dataset is partially processed and then a 2D class/3D map is used to re-pick the micrographs, the workflow should link back to the micrographs i.e. fork from the top of the diagram, rather than continuing in a linear arrangement.

The workflow diagrams have been updated (Extended Data Figs. 8-10). We have included the preprocessing and micrograph curation steps. We now also position forks in the diagram in line with the reviewer's suggestion.

27. Ext. Data Fig 9: What looks like a 2D-class side view is used for Topaz picking and combined with a separate picking job, but this is not explained in the text or legend. It's not clear why this was done here and not in the other workflows.

We have indicated in the updated workflow diagram (Extended Data Fig. 10) that a preferred orientation was identified by a streaking feature in the obtained map though the particle orientation distribution plot appeared to have sufficient angle sampling. The additional flexible loops and KDs in the holoenzyme sample might contribute to the difficulty of picking side views from template matching. We trained a separate Topaz model with the side view particles (confirmed by 2D averages). This approach successfully enriched the side views and produced isotropic reconstructions. In the hub datasets, we didn't detect the preferred orientation issue.

28. Supp. Videos – Indicate which video represents latent coordinate 1 or latent coordinate 2 as shown in Figure 3.

We have combined the movies into one movie (Supplementary Movie 2) and added captions to distinguish the two latent coordinates. The associated Figure in the main text has also been revised to present the motions in a better way.

Reviewer #2 (Remarks to the Author):

This interesting paper takes a three-pronged approach to examine the oligomerization state/stability of CaMKII holoenzymes. The three approaches are single-particle cryo-EM, molecular dynamics simulations and single-molecule fluorescence microscopy. The overall conclusion is a proposed assembly model where CaMKII goes through a dimer->tetramer intermediate on the path to a dodecameric or tetradecameric assembly. The cryo-EM studies appear well-done and show results consistent with other crystal structures and negative-stain single-particle reconstructions. Both dodecameric and tetradecameric assemblies of alpha and beta hubs (constructs lacking the kinase domain) and a full-length betaCaMKII were assessed. Interestingly, a form of the full-length betaCaMKII with an open hub structure was a large proportion of the particles and along with the mixtures of 12- or 14-mer hubs indicates an instability in the hub domain. The description of vertical and lateral domain interfaces is not new given there are higher resolution crystal structures available to draw these conclusions but is worthy of commenting on in the context of this paper. The resolution of the reconstructions is impressive and pushing the SPA with cryo-preserved specimens is an important advance, although there are some issues to address. The molecular dynamics simulation is a nice complement to the structural work, but don't seem they are dependent on the new data presented in this manuscript with one exception. The simulations of the open structure for betaCaMKII holoenzymes seems interesting, but the conclusion on the spreading of the instability seems speculative. The single-molecule fluorescence studies are also interesting and further support that there is instability in CaMKII. The results are complex and appear to depend heavily on model-based data analysis. While intermediates are clearly identified and mutations and nucleotide appear to alter the stability, the exact counting of molecules beyond a monomer/dimer seem difficult to accurately deduce. A general overall comment is that while the three approaches are complimentary, the manuscript, particularly the results, reads like three separate papers. There is also insufficient presentation/integration of a significant body of literature to judge the novelty/impact of the present manuscript.

We thank the reviewer for the appreciation of the cryo-EM structures, the complementary molecular dynamics simulations and single-molecule TIRF. We have extensively revised the manuscript to address the two overarching concerns regarding the integration of (i) the MD and TIRF with the cryo-EM and (ii) the present study with the previous literature. The revisions are spelt out in the responses below to the specific issues raised by this reviewer.

Critique:

Major

1. The primary data for the cryo-EM figures that leads to the SPA analysis is problematic. The images appear to be a tangled mass of protein and it's not clear what represents single particles on which all subsequent analysis is based. Better images of the primary data processed are needed with clearly identified single particles.

The images showed a high protein concentration of particles used in this study because 20 mg/ml is needed to preserve intact holoenzyme states. In addition, the Venus proteins are the major components of isolated hubs. These contribute to the density but were masked out of the subsequent analysis and 3D reconstructions. The problem is even more severe for the holoenzyme samples where both the Venus and the kinase domains dominate the density in the raw images but are not part of the 3D maps. As a result, we relied on blob picking, template picking, and Topaz picking, rather than visual identification. 2D classification right after blob picking, with the particle diameter as the only input, showed the clear 12-mer and 14-mer features. In all template-picking jobs, we were very cautious about template bias with all the input templates filtered to 20 Å (Extended Data Fig. 8,9). It would have been impossible for our cryo-EM structures to show such a high degree of consensus with the crystal structures (Extended Data Fig. 4e) if the particle picking was flawed.

2. Also, for the SPA, It is not clearly stated if each of the reconstructions comes from a single preparation of each construct, or if attempts were made to analyze multiple preparations of each purified protein. One important issue here is how consistent the distribution of D6 and D7 particles is across different preparations. Perhaps most importantly is whether the betaCaMKII holoenzymes where the open hub was the dominant (75%) form reproducible. Related is whether these D6 and D7 or open structures reported are the result of bacterial expression/protein purification. This does not seem considered anywhere in the manuscript. Compounding the concern is lack of indication of how reproducible the findings are between preparations.

We now state that the 14-mer and 12-mer reconstructions coexist in a single sample (Results Sections 1 and 4 - 1st paragraphs). The sample used for the near atomic

resolution constructions was the culmination of many experiments used to optimize concentration and minimize heterogeneity. The initial negative stain experiments were published almost 5 years ago (Khan et al., 2019). Fig. 3A of that study shows isolated holoenzymes interspersed with smaller assemblies initially thought to be proteolysis products. Subsequent experiments increasingly suggested that the heterogeneity might be due to disassembly; a hypothesis examined in this study.

We make no claims on the relative distributions of the D6 and D7 particles. Any claim based on the picked particle populations used in cryo-EM reconstructions is problematic, particularly in high-resolution reconstructions where the final datasets are a small fraction of the initial picked populations. See response to Reviewer-1 point-19.

From the heterogeneous refinement result, the open hub fraction (336,745 + 438,741 ptcls) in the final particle dataset of more than 2 million particles from Topaz is 37.5% (Extended Data Fig, 10). A large number of particles were not picked for the analysis due to the presence of factors listed in the response to Reviewer-1 point-19.

Of the eight high-resolution structures derived from X-ray crystallography reported thus far (Extended Data Fig. 4e), all except one (2F86.pdb) used a bacterial expression system. The exception used the insect sf9 expression system to report a structure that could not be distinguished from those obtained by bacterial expression. Subunit polymorphism was demonstrated previously by lower resolution EM studies. Two of these studies utilized bacterial expression. One used the insect expression system but docked an atomic model of a structure obtained by bacterial expression into the EM maps. Bacterial expression may be a concern; nevertheless, it has been used extensively for previous structural investigations.

3. Is there a reason that the assessment of the dodecameric CaMKII β preparations was abandoned (page 10 lines 204-205)? This data should be presented even if not achieving the resolution of the other reconstructions.

The image processing of the closed 12-mer hub present in the holoenzyme sample is now detailed in the workflow diagram (Extended Fig. 10) and the 8.4 Å resolution reconstruction shown in Fig. 5. See also response to Reviewer-1 point 19.

4. Clearer statements need made to the claims of resolution for the CaMKII β holoenzyme samples. It is not possible that the claimed resolutions are reflective of the entire holoenzyme as the KDs and then Venus tags were not resolved at all. The writing needs clarified here that the resolutions are specifically related to the hub domains and should identify exactly which residues the resolution claim applies to.

The reviewer is right. We now specify that the reconstructions are of the hub domains alone in all cases (see response to Reviewer-1 point-19). The unresolved KDs and Venus tags were masked out during FSC resolution estimation. We also note, as discussed in detail, the resolvability variations between specific residues reported as individual Q-scores (Figs. 2 and 7, Extended Data Fig. 1).

5. There is also no indication that the enzymatic activity of these preparations was assessed and whether the activity is consistent across different preparations. Some validation of the enzymatic activity of these preparations (when the KD is present) is essential.

The cryo-EM structures presented in this study are from enzymatically inactive samples through site mutagenesis. Three Venus tagged samples were studied, the alpha hub, the beta hub, and the beta holoenzyme with inactivating T286A, T306A and T307A residue substitutions. The reviewer's comment is relevant for the single molecule analyses of the native alpha and beta holoenzymes and the constitutively active beta T287D holoenzyme. The experiments on these constructs have been removed from this revised manuscript and moved to a separate manuscript which deals in depth with ATP-dependent KD interactions with calcium calmodulin (Khan et al., 2023). Western blot evidence for the calcium dependent autophosphorylation of the parent strains is shown in Extended Data Figure 5a.

6. This reviewer does not have the expertise to comment on the technical merits and findings from the MD simulations. The most intriguing observation appears to be the findings in Figure 4 where there is spreading of the instability shown by an increased B-factor. How consistent is this finding? Further, it does not appear the system has reached stability after 100 ns, how does this influence the interpretation of the results?

The Methods description of the MD simulations has now been expanded as stated in the response to the Editor. Supplementary Tables S3a and S3b describe the replicates performed for each CaMKII assembly and relate the present study to previous simulations respectively. The spreading of the instability shown by the increased B-factor associated with the further opening of the hub after dimer loss is seen in all three replicates (Fig. 7e-g).

The stable configuration would be complete disassembly. There was no expectation in the present simulation, or the earlier simulation on which it was modeled (Stratton et al., 2014), that disassembly would be complete in 100 ns. Our simulation was aimed to demonstrate further opening of the hub after dimer-mediated disassembly. Interestingly, episodic jumps were observed in all three replicates, They suggest, taken together with 3DFlex analysis of the single particle dataset, how cooperative dissociation of LC residue contacts could lead to further dimer disassembly.

Stability, as commonly understood for MD simulations, is the energy minimization and equilibration of the input structure before the production run. As now spelt out (Methods), this was achieved for all CaMKII assemblies simulated, including the open hub (plot in Supplementary Table S3a).

7. The description of the data concerning the single-molecule fluorescence is difficult to follow. There seems no need to summarize the work before presenting each of the analysis.

Point taken. The opening paragraph for this section has been removed and the entire section condensed to a couple of paragraphs (Results Section 3), Fig. 4 and Extended Data Figs. 5c-e. These retain only the data directly relevant to hub disassembly.

8. There is a similar challenge for fluorescence studies as noted for the SPA analyses. How were the preparations characterized for their enzymatic activity and how many separate preparations were assessed to evaluate the reproducibility of the findings?

See the response to point 5 regarding enzymatic activity. Protein purification for the cryo-EM and single molecule TIRF microscopy experiments was identical as now stated (Methods lines 561-563). The difference was in the subsequent sample preparation for each measurement. The preparation of the cryo-EM grids involved dilution of the sample to the targeted 20 mg/mL concentration and transfer to the Vitrobot chamber for plunge freezing. The process takes around one minute. For TIRFM experiments, the thawed samples were diluted 1000x. Preparation of the flow chambers (< 10 minutes) and subsequent acquisition of the video records (10 – 30 minutes) were at room temperature. Duplicate experiments were done for each separate sample/condition. Population size and F-test statistics for the different distributions obtained are listed in the associated legend (Fig. 4).

9. The experiment mostly directly assessing the stability of the oligomeric state is the aging experiment described in section C. However, the design is unclear. Was the preparation applied to the chip as single holoenzymes (or dissociated subunits) and then imaged at two different times, or diluted, aged, and then applied to the chips? Related, if the instability is as great as suggested why aren't the single bound molecules falling apart during the imaging protocol. Or are they and that explains some of the complex disappearance of fluorescent signals?

Correct. The aging experiment “most directly assesses the stability of the oligomeric state”. This experiment has been retained in the revised manuscript.

As noted in the response to the previous point-8, the samples for the cryo-EM and single-molecule TIRF measurements, as well as the newly added gel-filtration

experiments, were purified in an identical fashion from bacterial expression hosts. Samples from freshly prepared or frozen glycerol stocks thawed on ice were processed differently for the EM grids versus flow cells for TIRFM experiments. In the former case, the grid was plunge-frozen upon application of a <10-fold diluted sample (final 20 mg/mL concentration) within a minute. In the latter case, the samples were diluted, typically > 1000-fold, and incubated for 3 minutes for attachment to the GFP antibody-coated surface, then washed thrice to remove unbound CaMKII before placement on the TIRF microscope stage for observation for between 10-30 minutes. In the case of the aged (45-minute) samples, the flow cell was left at room temperature for 30 minutes prior to placement on the microscope stage.

Correct. The point of the comparison between the 15- and 45-minute samples was to demonstrate that the immobilized holoenzymes fall apart upon washout of unbound CaMKII due to the intrinsic lability of the hub. The gel filtration experiments complement the TIRFM.

10. The discussion of energies and hub stability in the discussion (lines 390 to 407) are convoluted and difficult to follow. If there are energetic penalties going from D7 to D6 then why would one ever see any significant fraction of 10-mers (D5?) noted in reference 18. Also in the middle of the same paragraph is a passing note about proteolysis that might influence the analyses. This might be true, but then an assessment of the state of the molecules under analysis using orthogonal techniques seems warranted.

11. Why do the authors dismiss specific residue differences in the hub domains leading to differences in the final oligomeric structures? Lines 384-386? There is a passing comment about crystallization conditions being an issue that makes little sense.

12. The thought expressed in lines 386-389 is additionally unclear.

13. Line 418 suggests that the present study “supplants” earlier ideas. I do not understand this claim. The study is based on, and modestly extends many earlier studies. The other points in the conclusionary paragraph seem a collection of not well-developed thoughts about LTP, local protein synthesis, role of the R peptide...etc. Overall, the Discussion needs a major rework to improve the integration of the findings in this paper from earlier literature.

The Discussion section of the manuscript has been extensively reworked to improve clarity and resolve the overarching concern of this reviewer regarding insufficient integration with the literature to permit an evaluation of the novelty of the present work. Specific responses to points 10-13 are noted below.

10. *We did not resolve any 10-mer ring assemblies in our study and concur with this reviewer that the formation of smaller holoenzymes could be unfavorable (Fig. 8 legend). The concern regarding linker proteolysis pertained to the previous freeze-etch replica study that counted kinase domains, rather than hub association domains, to*

report 10-mer β holoenzymes (Kanaseki et al., 1991). Proteolysis has been previously reported by us and others to occur under certain conditions, but the freeze-etch study predates these observations and there is no hint that the authors were aware of this concern.

11. Lines 384-86 (initial submission) refer to published crystal structures of the human δ and γ hubs. We and others have shown that α and β hub polymorphs coexist in EM preparations. The residues and energetics of the lateral contact interfaces are conserved between all mammalian isoforms (Extended Data Fig. 4). It is likely, therefore, that polymorphism is intrinsic to all isoform hubs but has not been demonstrated for the δ and γ isoforms as they have not yet been studied by single-particle cryo-EM. The point we make about crystallization conditions is simply that, as noted by reviewer-1, “the dependence of X-ray crystallography on homogenous populations for crystallization obscures heterogeneity”. The merit of using cryo-EM structural determination is its ability to sort out heterogeneous particles in a single biochemical preparation and derive their structures.

12. Lines 386-389 (initial submission) cite an idea expressed in an earlier publication that “the oligomerization states of various CaMKII assemblies might be different” so that “resultant differences in the packing of kinase domains tune the response of variant forms of CaMKII to calcium signals”(Hoelz et al., 2003). As now stated in the final Discussion paragraph, this idea is unlikely to be correct because we think the coexistence of hub polymorphs is common to all mammalian isoforms as explained in the response to point-11.

13. We certainly did not intend to give the impression that this work “supplants” earlier work. This study, like any other, would not have been possible in the absence of many earlier studies. We have restructured the manuscript, in particular the Results and Discussion sections, to cite earlier work as appropriate.

What is challenged are some speculative ideas, not the experiments, advanced in earlier publications. Conversely, other ideas are supported – importantly the ideas for the molecular basis of strain in hub assemblies and hub assembly/disassembly by the serial addition/loss of dimers. We also note the equivalence of the α and β isoform inter-domain hub contacts as support for speculation made long ago regarding the role of $\alpha\beta$ hetero-oligomers in cellular localization but emphasize that extrapolation from in vitro data to in vivo function is difficult and that this idea remains to be tested (final Discussion paragraph). We hope that the revised manuscript will allow a better appreciation of our characterization of the intrinsic hub lability in the context of the relevant literature.

Minor

1. Page 10, line 196 – the mutations do not eliminate endogenous phosphorylation. The eliminate phosphorylation of specifically those sites.

Thank you. The phrase “endogenous phosphorylation” has been deleted.

2. Page 18-19 lines 383-384 the finding of the gamma and delta isoforms are reversed. In the Rellos et al. paper, gamma is D6, and delta is D7.

Thank you. This error has been corrected (Extended Data Fig. 4e).

Reviewer #3 (Remarks to the Author):

The authors investigated the structure and stoichiometry of CaMKII beta hub domain and holoenzyme using cryoEM, MD simulations, and single molecule fluorescence microscopy. CaMKIIalpha hub structures (fused to Venus) are reported using cryoEM, and they show both 12mer and 14mer stoichiometries. They simulate 14mer CaMKII alpha structure to compare the two interfaces that hold the hub together. CaMKII beta holoenzyme structures (fused to Venus) are reported using cryoEM, and interestingly they observe open ringed structures, which are further analyzed using MD. CaMKII beta holoenzyme fused to Venus is analyzed in bulk using single molecule TIRF paired with photobleaching. The most interesting and novel result is the open-ringed structure, but more details need to be provided about the sample itself as well as the data analysis. It also would have been nice to show this in a tag-less construct. The other results muddy this potentially exciting finding – especially the smTIRF, which seems out of place.

We appreciate the interest in, and appreciation of the novelty of the open structure in the overview given by this reviewer. More details are provided in the revision, as noted above in the responses to reviewer-1 and reviewer-2, and below to specific issues raised by this reviewer. As regards other points in the overview

(a) The flexibility of the open structures is analyzed in two ways, 3DFlex refinement of the single particle dataset and the MD.

(b) All constructs, not just the holoenzymes, were fused to Venus. The TIRFM experiments would not have been possible without the tag, and it also facilitated the isolation and analysis of the assemblies by gel filtration. The published mutagenesis analysis of the lateral contact interface used the Venus-tagged α isoform (Sarkar et al., 2017), repeated for the corresponding Venus-tagged β isoform in our study. We checked that the Venus-tagged parent strains were competent for primary autophosphorylation.

(c) More detail is now provided in the Methods, the associated Extended Data Figure legends and Supplementary Material Tables.

(d) We do not resolve Venus in our 3D-maps, the comparison of our α 12-mer and 14-mer maps shows good agreement with the published hub crystal structures, and the

calcium calmodulin dependence of kinase activity has been shown to match results obtained for tag-less constructs (Sarkar et al., 2017).

(e) We agree that some of the TIRM experiments included in the initial publication were out of place. These have been removed to another manuscript (Khan et al., 2023). The TIRFM experiments that demonstrate the intrinsic lability of the holoenzyme with time and concentration have been retained. We also kept the experiments on the single residue mutations that are important to establish that the disassembly is due to dimer loss, as well as to demonstrate the conservation of the lateral contact interface between the α and β hubs.

Main concerns

1. The authors use a wide repertoire of techniques to characterize CaMKII. However, as presented, the results do not flow well. The single molecule TIRF experiment is interesting on its own, but it is unclear how it's related to the cryoEM results. The dimer mutations come out of nowhere, as does monitoring activation (by adding ATP).

We agree that the results did not flow well in the initial submission. Thank you for this critique. We have restructured the presentation extensively to address this issue.

The dimer mutations validate the conservation of the lateral contact interface between the α and β hubs obtained from bioinformatics. Additional experiments have been included to show that the β F458A mutation disassembles isolated hubs as well as holoenzymes (Extended Data Fig. 4g). These results form an integral part of the overview of the hub structures (Extended Data Fig. 4e, f). They set the stage for experiments where this mutation is used to calibrate dimer loss during holoenzyme disassembly (Fig. 4).

The TIRFM experiments on the ATP dependence have been moved to a separate manuscript.

2. The writing is also disjointed and incorrect at times. For example, in the abstract they write "hub subunit stoichiometry is important for kinase activity" – but it's unclear what they mean by this. The kinase is activated well even when the hub is not present. I also find the title misleading as no energetics are performed in this paper.

Point taken. The clarity has hopefully been improved in the revised version. We did not mean to imply that the hub is required for kinase activity. The sentence in question has been revised and the appropriate citation (Sloutsky et al., 2020) given in the Introduction (3rd paragraph) to make its meaning clear.

An energetic analysis of all hub structures is presented in Extended Data Fig. 4e-f. The MD comparison of the flexibility of the extracted tetramers with their parent hubs

(Fig. 3d-f) is a measure of the energy cost of the steric strain imposed by ring closure in the 12-mer and 14-mer hubs.

3. The model presented in Figure 6 is problematic for several reasons. One is that the comparisons being made in the manuscript are of different variants and different fusion constructs. For instance, hub versus holoenzyme, and both are fused to a large fluorescent protein which may impact population size. Second, cryoEM is a fantastic tool for generating structures, but is not always representative of the populations in solution for technical reasons. Thus, it is inappropriate to convert these populations into an energetic argument. As a side note, showing a structure of D8 here is misleading as that was not solved in this manuscript.

We regret that the model was misunderstood by this reviewer due to the unclear presentation. The model is now explained in more detail in the associated text as well as its legends. We address the misconceptions below and then give a condensed explanation of the model. We trust this explanation, together with the revised manuscript text is satisfactory.

(a) The model is a generic model that is not restricted to the results of the present study. Instead, it is based on the integration of this study with all published near atomic hub structures.

(b) The model considers only the intrinsic stability of the hub and not interactions with associated structural elements or allosteric modulators. There is no evidence that Venus alters hub structure or dynamics (response to overview point-d).

(c) An attempt to convert populations based on cryo-EM particle numbers into an energetic argument is not made anywhere in the manuscript because the cryo-EM image processing steps eliminated a lot of particles based on the algorithms used. Since there is not sufficient investigation in the cryo-EM field about the existence of bad particles either in protein purification or vitrification steps, it is premature to draw any direct correlation of particle numbers with energy estimation in the MD simulations or any kinetics inference (see response to Reviewer-1 point-19).

(d) As should be clear from (a), it is appropriate to present a structure reported in another study, but the study should have been cited. It is now properly cited (Buonarati et al., 2021). Our model brings together two concepts based on physical chemistry. First, ring formation from flexible building blocks, such as proteins, has both an enthalpic and entropic cost due to the steric constraint. Second, serial extension of linear polymers depends on the building block concentration. Previous studies have documented the strain in closed hubs (Stratton et al., 2014). Our MD simulations show further that the intrinsic curvature of the unconstrained tetramer building blocks is intermediate between the curvature of the 12-mer and 14-mer hubs. It follows that the 12-mer and 14-mer hubs are roughly isoenergetic and that the strain will increase for hubs with higher and lower stoichiometries. Previous studies have shown that in open metazoan hubs,

serial extension is a concentration of the dimer building blocks (Bhattacharyya et al., 2016). The structure and analysis of the open hub structure together with the TIRFM experiments extend this observation and generalize the physically-sound argument. The predictions of the model will, of course, need to be tested.

4. It's quite surprising that so many particles of CaMKII beta holoenzyme are in an open conformation. Why are the authors adding 4 mM Mg²⁺ to these samples? Since this result is unexpected, the authors should show some validation that their sample is behaving well in solution under these conditions before adding to the grid. (Mass photometry, SEC-MALS, or something similar). It would be nice to show raw images in the supplement as well.

It is not possible from the number of particles used for the 3D reconstruction of any particular structure, in this case, the open hub structure, to estimate its population fraction (see responses to Reviewer-1 point-19 and point-3c (this reviewer)). It is likely to be a minority species, in addition to the smaller assemblies suggested by the TIRFM and the new gel filtration data.

We now clarify in Methods, that Mg²⁺ was in the buffer used for the TIRFM experiments alone. Mg²⁺ is an essential cofactor for ATP hydrolysis. The TIRFM experiments retained in our manuscript were part of a larger study that investigated in detail the ATP dependence of calmodulin – CaMKII interactions (Khan et al., 2023).

Analytical gel filtration experiments to support the disassembly at room temperature seen in the TIRFM experiments are now included in the revised manuscript (Extended Data Fig. 5b). Raw cryo-EM images are shown for all samples, but the crowding makes it difficult to readily characterize the greater heterogeneity of the holoenzyme sample relative to the isolated hubs. The 2D-class averages of the structures are more informative (see response to Reviewer-2 point-1).

5. In the smTIRF experiment, the authors need to show representative images of their single molecules. They also need to clarify the conditions used in all these samples, as the buffer components are not explicitly stated everywhere (which of them have ATP/Mg, etc). The starting intensity of the dimer is ~300, whereas the starting intensity of the holoenzyme is >4000. Based on the dimer intensity, this would indicate that each Venus is ~150 and a 12mer should be 1800, and a 14mer should be 2100. The authors need to explain why the holoenzyme intensity is so much greater.

A representative image is part of the workflow for processing TIRFM data records now described in Extended Data Fig. 7. Extensive details and further images are documented in (Khan et al., 2023) and on GitHub.

More details on sample preparation are now given in Methods. Initial protein expression and isolation were common to all experiments, but the TIRFM and gel-

filtration experiments were at room temperature (> 30 minutes duration) in contrast to the preparation and observation for cryo-EM. The buffer components are now explicitly stated (Methods -> Protein preparation (2nd paragraph). None of the experiments retained in this revised manuscript used buffers with Mg²⁺.

The difference between the dimer and holoenzyme photobleaching curves (Extended Data Fig.5d) is reproducible for the complete step populations. The mean step size for the dimer population is 199±60 counts/step, while it is 317±130 counts/step for holoenzymes (Extended Data Fig.5e). Both Figures were shown in the initial submission, but the difference was not well explained. The difference between the expected cumulative step intensity (200x14=2786(±224) counts) and the initial intensities in the holoenzyme records shown is due to the residual autofluorescence contribution. The residual autofluorescence and steps missed in the initial segments of the video records account for the observed cumulative holoenzyme step intensity (317x14=4438(±486) counts), as now noted in the legend to Extended Data Fig.5e.

Minor points

-the data shown in Figure 2 is not so novel – this has been pointed out many times before in structures and MD

It is helpful to review this Figure (now Fig. 3) panel-by-panel to evaluate novelty. Panels a and b compare the relative strengths of the lateral and vertical contact interfaces, with MD-derived B-factors, with the results in line with earlier evidence, that included MD (Stratton et al., 2014). The purpose of these panels here is to validate the current MD protocol for the subsequent comparison with the Q-score (Panel c), a recently developed metric for cryo-EM map resolvability (Pintilie et al., 2020). This comparison is novel in the context of the CaMKII literature. Panels d-f show that when tetramer sub-assemblies are extracted from closed 12-mer and 14mer hubs they relax to superimposable conformational distributions with broad spread and mean angular curvature between the values for the two hubs. This analysis, as far as we are aware, has not been done before. The result is critical to our assembly model that seeks to explain the coexistence of 12-mer and 14-mer polymorphs (Figure 8).

There are not “many” MD simulations of hub structures reported in the literature. We now provide an index of previously reported simulations that we are aware of to better place the simulations presented in this study in the context of the literature (Supplementary Table S3b). The software packages for full-atom simulations, such as those presented here, largely overlap. So, the studies are distinguished from each other not by the trajectories obtained per se, but by their subsequent analyses that address different issues. While the simulations in Figure 3. d-f are novel, a similar simulation (Stratton et al., 2014) preceded our open hub simulations (Figure 7. d-g). An

experimental open hub structure was not available for comparison with this earlier simulation, and our simulations have been analyzed in greater spatiotemporal detail.

-different fonts and font sizes are used

We apologize for the inconsistencies. The revised manuscript has been checked to ensure that such occurrences have been eliminated.

-in Fig 4C, one subunit is colored white to indicate an open ring? Is this on purpose?

Yes, it is on purpose. The Figure follows a Figure where the same coloring scheme was used for the comparable simulation (Stratton et al., 2014).

CITATIONS

- Bhattacharyya, M., M.M. Stratton, C.C. Going, E.D. McSpadden, Y. Huang, A.C. Susa, A. Elleman, Y.M. Cao, N. Pappireddi, P. Burkhardt, C.L. Gee, T. Barros, H. Schulman, E.R. Williams, and J. Kuriyan. 2016. Molecular mechanism of activation-triggered subunit exchange in Ca(2+)/calmodulin-dependent protein kinase II. *Elife*. 5.
- Buonarati, O.R., A.P. Miller, S.J. Coultrap, K.U. Bayer, and S.L. Reichow. 2021. Conserved and divergent features of neuronal CaMKII holoenzyme structure, function, and high-order assembly. *Cell Rep*. 37:110168.
- Hoelz, A., A.C. Nairn, and J. Kuriyan. 2003. Crystal structure of a tetradecameric assembly of the association domain of Ca2+/calmodulin-dependent kinase II. *Molecular cell*. 11:1241-1251.
- Kanaseki, T., Y. Ikeuchi, H. Sugiura, and T. Yamauchi. 1991. Structural features of Ca2+/calmodulin-dependent protein kinase II revealed by electron microscopy. *J Cell Biol*. 115:1049-1060.
- Karandur, D., M. Bhattacharyya, Z. Xia, Y.K. Lee, S. Muratcioglu, D. McAfee, E.D. McSpadden, B. Qiu, J.T. Groves, E.R. Williams, and J. Kuriyan. 2020. Breakage of the oligomeric CaMKII hub by the regulatory segment of the kinase. *Elife*. 9.
- Khan, S., K.H. Downing, and J.E. Molloy. 2019. Architectural Dynamics of CaMKII-Actin Networks. *Biophys J*. 116:104-119.
- Khan, S., J.E. Molloy, H. Puhl, H. Schulman, and S.V. Vogel. 2023. Real time single molecule imaging of CaMKII calmodulin interactions. *Biophysical Journal*. Under Review.
- Pintilie, G., K. Zhang, Z. Su, S. Li, M.F. Schmid, and W. Chiu. 2020. Measurement of atom resolvability in cryo-EM maps with Q-scores. *Nat Methods*. 17:328-334.
- Sarkar, P., K.A. Davis, H.L. Puhl, 3rd, J.V. Veetil, T.A. Nguyen, and S.S. Vogel. 2017. Deciphering CaMKII Multimerization Using Fluorescence Correlation Spectroscopy and Homo-FRET Analysis. *Biophys J*. 112:1270-1281.
- Sloutsky, R., N. Dziedzic, M.J. Dunn, R.M. Bates, A.P. Torres-Ocampo, S. Boopathy, B. Page, J.G. Weeks, L.H. Chao, and M.M. Stratton. 2020. Heterogeneity in human hippocampal CaMKII transcripts reveals allosteric hub-dependent regulation. *Sci Signal*. 13.
- Stratton, M., I.H. Lee, M. Bhattacharyya, S.M. Christensen, L.H. Chao, H. Schulman, J.T. Groves, and J. Kuriyan. 2014. Activation-triggered subunit exchange between CaMKII holoenzymes facilitates the spread of kinase activity. *Elife*. 3:e01610.

Reviewers' comments:

Reviewer #1 (Remarks to the Author):

The revised manuscript is greatly improved and the authors have addressed all my concerns and provided adequate explanation in the rebuttal letter and manuscript. The new and revised figures have added important detail to the decisions made during cryo-EM reconstruction and the fit of the models to the maps. I agree that the extraction of some of the results into another manuscript has streamlined and improved the flow of this manuscript. I have a few minor corrections and suggestions:

1. Figure 1 – The dashed hexagon is not explained in the legend

2. Rebuttal point 6 “Furthermore, we improved our open-ring structure by refining the model against the unsharpened map. The original deposited model was refined against the sharpened map. The opening region, with lower resolution, was oversharpened in this map. As a result, the density becomes fragmented and difficult to interpret. In contrast, the unsharpened map shows clear and connected densities that are easier for secondary structure identification. Our updated model refined against the unsharpened map is more accurate which contributed to the improved atom inclusion level of 82%.”

I agree that the refined model of the open-ring structure fits the map much better than before, as reflected in the updated validation report, especially the domains closest to the ring opening. For the pathology you have described of the sharpened map, I would recommend a local sharpening approach (e.g. DeepEMhancer - CryoSPARC, LocalDeblur - Scipion) that locally estimates B-factors and sharpens segments of the map accordingly. In my experience, this improves the interpretability of the map at a consistent threshold. It may not change the model but will likely result in a nicer sharpened map for deposition and figures.

3. Rebuttal point 17 “We have replaced the map correlation coefficient with model RMSD values for a more rigorous structural comparison...”

I agree this is more robust, however, you must clarify for each RMSD value whether this was calculated on an all-atom basis (c.f. main-chain or c-alpha atoms only) and how many residues were compared. Most software packages automatically exclude atoms/residues according to distance and sequence alignment criteria by default when performing RMSD calculations.

4. Rebuttal point 20 “We have checked that the axes labels are preserved in the revised Fig. 4 and the associated Extended Data Figs. 5c-e.”

Ext Data Fig 5c still has cropping of the left axis labels.

Reviewer #2 (Remarks to the Author):

I have some reservations regarding the execution and presentation of the MD simulations in the manuscript. Firstly, the duration of 50 - 100 ns falls short of the current state of the art in the field. This limitation is compounded by the likelihood of slow motions leading to the opening of the multimer, making the timeframe insufficient for capturing significant structural transitions. Secondly, certain technical approaches appear unconventional; for instance, computing a "B factor from RMSF" instead of directly plotting RMSF, which is the standard practice, raises questions. Furthermore, the absence of convincing evidence supporting the authors' propositions is notable, particularly concerning the observed increase in RMSD. Additionally, the RMSF profiles, particularly the RMSF-based B-factors, seem to primarily reflect localized dynamics occurring on the nanosecond timescale, without offering substantial insight into the observed opening phenomenon. While angle fluctuations provide some rationale, their significance within the short simulation timeframe remains

questionable.

Considering these limitations, it might be beneficial for the authors to incorporate a control group within their simulations, which should remain stable. Comparing the effects observed over short time windows to this control could potentially reveal significant deviations worthy of exploration. Moreover, I strongly advocate for longer simulation durations, extending to several hundred nanoseconds per replica per system, to better capture dynamic processes.

On a more positive note, I believe that these concerns regarding the MD simulations do not diminish the value of the experimental findings presented in the manuscript. It may be prudent to consider rearranging the manuscript to prioritize experimental results, relegating MD simulation outcomes to a supporting role, or mentioning only key findings if deemed necessary. The wealth of insights derived from the experimental structures outweighs the limitations encountered in the MD simulations.

Reviewer #3 (Remarks to the Author):

Thanks to the authors for addressing reviewer concerns. The paper reads much more smoothly.

Reviewer #1:

The revised manuscript is greatly improved, and the authors have addressed all my concerns and provided adequate explanation in the rebuttal letter and manuscript. The new and revised figures have added important detail to the decisions made during cryo-EM reconstruction and the fit of the models to the maps. I agree that the extraction of some of the results into another manuscript has streamlined and improved the flow of this manuscript. I have a few minor corrections and suggestions:

We thank this reviewer for the positive comments. We have benefited from the guidance provided for the revision of our initial submission.

1. Figure 1 – The dashed hexagon is not explained in the legend

The dashed hexagon denotes the hub, as now noted in the Figure 1 legend.

2. Rebuttal point 6 “Furthermore, we improved our open-ring structure by refining the model against the unsharpened map. The original deposited model was refined against the sharpened map. The opening region, with lower resolution, was oversharpened in this map. As a result, the density becomes fragmented and difficult to interpret. In contrast, the unsharpened map shows clear and connected densities that are easier for secondary structure identification. Our updated model refined against the unsharpened map is more accurate which contributed to the improved atom inclusion level of 82%.”

I agree that the refined model of the open-ring structure fits the map much better than before, as reflected in the updated validation report, especially the domains closest to the ring opening. For the pathology you have described of the sharpened map, I would recommend a local sharpening approach (e.g. DeepEMhancer - CryoSPARC, LocalDeblur - Scipion) that locally estimates B-factors and sharpens segments of the map accordingly. In my experience, this improves the interpretability of the map at a consistent threshold. It may not change the model but will likely result in a nicer sharpened map for deposition and figures.

We tested DeepEMhancer to locally sharpen our open ring map, and indeed it provides a better sharpened map. We replaced the original Figure 5b (now Figure 6b) with the DeepEMhancer map and specified how we post-processed the map in the figure legend. The model remained unchanged as the model building is still limited by the local resolution of the map.

3. Rebuttal point 17 “We have replaced the map correlation coefficient with model RMSD values for a more rigorous structural comparison...”

I agree this is more robust, however, you must clarify for each RMSD value whether this was calculated on an all-atom basis (c.f. main-chain or c-alpha atoms only) and how many residues were compared. Most software packages automatically exclude atoms/residues according to distance and sequence alignment criteria by default when performing RMSD calculations.

Thanks for the suggestion. We have included more details in RMSD calculation in the Materials and Methods Section – Image Processing and Model Refinement. The values reported

here were obtained from MatchMaker with all default settings. The RMSD comparisons of the alpha hub and beta hub (Results Section-1, paragraph 2) were calculated on 129 (out of all 133) pruned C α pairs.

4. Rebuttal point 20 “We have checked that the axes labels are preserved in the revised Fig. 4 and the associated Extended Data Figs. 5c-e.”

Ext Data Fig 5c still has cropping of the left axis labels.

Thank you. The error has been corrected.

Reviewer #2:

I have some reservations regarding the execution and presentation of the MD simulations in the manuscript.

1. Firstly, the duration of 50 - 100 ns falls short of the current state of the art in the field. This limitation is compounded by the likelihood of slow motions leading to the opening of the multimer, making the timeframe insufficient for capturing significant structural transitions.

The duration of the MD simulations is set by their scope which is now stated more precisely (lines 152-54, 306-307). The overarching aim is to provide a supporting chemical rationale for the local dynamics revealed by the resolvability and flexibility analyses of the cryo-EM structures. The simulations are presented as two, distinct subsets.

The first subset (Results Section-2) evaluates the match between the RMSF and Q-scores for the interfacial contact residues and, based on the satisfactory match, further explores the intrinsic flexibility of these contacts in an open tetramer sub-assembly unlocked from the closed ring holoenzyme (Fig.4). The match with the Q-scores and the agreement between the angular distributions obtained from independent replicates implies that the duration of the simulations is sufficient to achieve steady-state. This is consistent with chemical intuition that sidechain motions and interdomain hinge motion occur on picosecond and nanosecond timescales respectively.

The second subset (Results Section-4 final paragraph) was motivated by our Q-score resolvability and 3D-flex analysis of the 12-mer open hub structure. In addition, while this structure has not been reported before, its existence was proposed for subunit exchange supported by an analogous MD simulation (Stratton et al., 2014). It was important that the simulation be repeated and refined based on the new structural data. We found that the hub opens, consistent with the previous simulation, and the match with our experimental data revealed interesting details. That said, since hub disassembly is a slow process, we did not expect to achieve a steady state. Thus, these simulations, most categorically, do not resolve the mechanism of dimer dissociation beyond the clues obtained from the structural analysis or even show it is the predominant route for hub disassembly, as this reviewer may have presumed. This issue is an important one that we may address in the future. It is not addressed in the present work. Accordingly, in compliance with the suggestion of this reviewer, we have moved these simulations to Extended Data Fig. 6. We have further shortened and merged the relevant text with the previous section on the flexibility analysis of the open hub dataset to more closely link the two.

2. Secondly, certain technical approaches appear unconventional; for instance, computing a "B factor from RMSF" instead of directly plotting RMSF, which is the standard practice, raises questions.

We agree that the presentation of RMSF values is the more conventional approach. The RMSF values are now plotted directly (Fig. 4, Extended Data Fig. 6) instead of the B-factor (B-fac) values derived from them. We should point out that the B-fac conversion ($B_fac = (8/3) \pi^2 (RMSF^2)$) is a standard option for RMSF calculations in Gromacs 2018 and later versions, precisely to enable comparison with structural metrics such as the Q-score. We therefore list both RMSF and B-fac correlations with the Q-score (Results Section-2, 1st paragraph) and state the relation between them (Materials and Methods – Molecular Dynamics); leaving it to the readers to select the option with which they are more comfortable.

3. Furthermore, the absence of convincing evidence supporting the authors' propositions is notable, particularly concerning the observed increase in RMSD. Additionally, the RMSF profiles, particularly the RMSF-based B-factors, seem to primarily reflect localized dynamics occurring on the nanosecond timescale, without offering substantial insight into the observed opening phenomenon. While angle fluctuations provide some rationale, their significance within the short simulation timeframe remains questionable.

Control simulations on closed hubs (Extended Data Fig.6 (old Fig. 7)) were run to establish the observed increases in RMSD in the open hub were significant (detailed in response to point-4 below). The reviewer is correct that these increases reflect localized dynamics (see response to point-1). We also agree that these simulations do not provide insight into the key step of dimer dissociation that causes the hub to open, as now clarified (Results final paragraph). Dimer dissociation events, as reported by the TIRF experiments, occur on the second-to-minute timescale. Simulation of these events would be challenging even with "state of the art" facilities as detailed in the response to point-5 below.

4. Considering these limitations, it might be beneficial for the authors to incorporate a control group within their simulations, which should remain stable. Comparing the effects observed over short time windows to this control could potentially reveal significant deviations worthy of exploration.

Short (20 ns) control simulations of closed 14-mer and 12-mer hubs were conducted (Extended Data Fig. 6a,b). The separation across a dimer in these closed hubs was measured using the same angle as that used to measure the change in the separation for the open hub structure engineered in silico from the closed 14-mer. The angular separation in all 3 replicates of the closed 14-mer or 12-mer hubs remains "stable". The separation deviated significantly in the open hub within 10 ns relative to the angle variations (Extended Data Fig. 6e). Exploration of these deviations revealed episodic jumps in all 3 100 ns open hub time trajectories. These events were consistent with the hinge-like behavior of the lateral interfaces suggested by the 3D-Flex analysis (Fig. 7 (old Fig. 6)).

5. Moreover, I strongly advocate for longer simulation durations, extending to several hundred nanoseconds per replica per system, to better capture dynamic processes.

This critique is related to the overarching concern (point-1) of this reviewer. It has been dealt with in detail in our response to point-1. Our simulations do not model or explain the long-term dynamics of dimer dissociation reported by the TIRF experiments. Long simulations for this purpose would need to be on the order of tens of microseconds for an incremental gain, at

best, in knowledge. The effort needed can be assessed from recent simulations to study the docking of the KD regulatory segment on alpha hubs (Karandur et al., 2020). These simulations, run over several microseconds on the Anton2 supercomputer as documented in Table S3b, still could not capture dimer dissociation.

6. On a more positive note, I believe that these concerns regarding the MD simulations do not diminish the value of the experimental findings presented in the manuscript. It may be prudent to consider rearranging the manuscript to prioritize experimental results, relegating MD simulation outcomes to a supporting role, or mentioning only key findings if deemed necessary. The wealth of insights derived from the experimental structures outweighs the limitations encountered in the MD simulations.

We value the appreciation of our work by this reviewer and have implemented the advice to limit the description of the MD simulations to one more compatible with their supporting role, as detailed above.

Reviewer #3:

1. Thanks to the authors for addressing reviewer concerns. The paper reads much more smoothly.

Thank you

Karandur, D., M. Bhattacharyya, Z. Xia, Y.K. Lee, S. Muratcioglu, D. McAfee, E.D. McSpadden, B. Qiu, J.T. Groves, E.R. Williams, and J. Kuriyan. 2020. Breakage of the oligomeric CaMKII hub by the regulatory segment of the kinase. *Elife*. 9.

Stratton, M., I.H. Lee, M. Bhattacharyya, S.M. Christensen, L.H. Chao, H. Schulman, J.T. Groves, and J. Kuriyan. 2014. Activation-triggered subunit exchange between CaMKII holoenzymes facilitates the spread of kinase activity. *Elife*. 3:e01610.